



# Assessment of Operational Non-Time Critical Sentinel-6A Michael Freilich Radio Occultation Data: Insights into Tropospheric GNSS Signal Cutoff Strategies and Processor Improvements

Saverio Paolella[1], Axel Von Engeln[1], Sebastiano Padovan[1], Riccardo Notarpietro[1], Christian Marquardt[1], Francisco Sancho[1], Veronica Rivas Boscan[1], Nicolas Morew[1], and Francisco Martin Alemany[1]

[1]EUMETSAT, Darmstadt, Germany

**Correspondence:** Saverio Paolella (Saverio.Paolella@external.eumetsat.int)

**Abstract.**

This study presents an exhaustive assessment of the Sentinel-6A Michael Freilich Radio Occultation (RO) data, focusing on the evaluation of bending angle products derived from the EUMETSAT-provided RO Non-Time Critical (RO-NTC) data collected between September and December 2021.

This analysis confirms the satellite's capability to exceed its target of 770 quality checked bending angle profiles per day, with an availability rate of 99.9%, demonstrating the robustness of the mission's operational performance. A detailed examination of the Signal-To-Noise Ratio (SNR) and phase noise indicates the high-quality nature of the data. The study also analyses the benefits of employing SNR-based signal cutoff strategies and L2 signal extrapolation in the troposphere, where it is more susceptible to SNR reductions. Furthermore, the paper details some processor enhancements, which led to improved bending

angle statistics, particularly below 22 km altitude. Additionally, the analysis revealed terrestrial interference signals on the L2 frequency, confirming that they do not significantly compromise the Sentinel-6A RO data quality.

The validation of the EUMETSAT processed Sentinel-6A RO-NTC data against the European Centre for Medium-Range Weather Forecasts (ECMWF) short-range forecasts and comparisons with Metop-B/C and EUMETSAT-processed SPIRE occultations, highlights the reduction in random error and modifications in the tropospheric bias structure, a result of the enhance-

ments in data processing techniques. This comprehensive analysis confirms the high quality of the EUMETSAT Sentinel-6A bending angle products and underlines the satellite's contribution to the EUMETSAT legacy of precise and reliable radio occultation data for weather forecasting and climate research.

## 1 Introduction

The Sentinel-6A satellite, also known as Sentinel-6 Michael Freilich, represents a significant milestone in the ongoing effort

to monitor the level of Earth's oceans with altimeters. Launched on November 21$^{st}$, 2020, from Vandenberg Air Force Base in California, it marks the continuation of a three-decade legacy of sea-surface height measurement that began with the TOPEX/-Poseidon mission and was carried forward by the Jason series of satellites Jason 1, 2, and 3 (Donlon et al., 2021b; Jiang et al.,


2023). Notably, Sentinel-6A is part of the Jason-CS (Continuity of Service) program, aimed at ensuring the uninterrupted collection of critical oceanographic data.

Scheduled to be joined by its sibling, Sentinel-6B, in 2025, Sentinel-6A mission is not limited to altimetry. It also includes a GNSS RO instrument called TriG (peng Ho et al., 2020), developed in collaboration between the National Aeronautics and Space Administration (NASA) and Jet Propulsion Laboratory (JPL). The RO technique (Melbourne et al., 1994; Kursinski et al., 1997, 2001; Hajj et al., 2002; Jakowski et al., 2009) is a powerful method used to measure the Earth's atmospheric properties, including temperature, pressure, and humidity profiles, as well as ionospheric electron density. This technique

involves the transmission of radio waves from a Global Navigation Satellite System (GNSS) satellite, which are then received by a low Earth orbiting (LEO) satellite, such as Sentinel-6A. As these radio waves pass through the Earth's atmosphere, they are refracted due to the varying density and ionization levels of the atmospheric layers. The amount of this refraction provides valuable information about the atmospheric conditions along the signal's path. By precisely measuring the changes in the phase and amplitude of the received signals, one can infer the bending angle profile of the radio waves as they traverse the Earth's

atmosphere. This bending angle, which is the main output of the EUMETSAT RO processors, is an important parameter that, when processed through the Abel transform (Fjeldbo et al., 1971), yields profiles of atmospheric refraction index, which in turns depends on temperature, pressure, and humidity (Smith and Weintraub, 1953) at different altitudes, as well as electron density in the ionosphere.

    The RO technique offers several advantages over traditional atmospheric sounding methods. It provides global coverage,

particularly when the instrument flies on polar orbits, including remote oceans and polar regions where in-situ measurements are poor or unavailable. RO measurements are not significantly affected by cloud cover or precipitation, allowing for consistent and reliable atmospheric profiling under various weather conditions. A distinct feature of the RO technique is its measurement of time, which inherently requires no calibration, thus making it a stable reference or anchor for other measurement techniques. This attribute enhances the reliability and consistency of the data obtained. The Sentinel-6A satellite, equipped with a state-of-

the-art RO instrument, represents a continuation and enhancement of this observational capability, ensuring the availability of high-quality atmospheric data for weather forecasting, climate monitoring, and scientific research (Anthes, 2011; Steiner et al., 2001; Yen et al., 2010).

    The Sentinel-6 RO receiver collects GNSS signals through two occultation antennas positioned in both the velocity and anti-velocity directions, for tracking rising and setting occultations respectively. The receiver is equipped also with a zenith-looking

antenna dedicated to track signals for the precise orbit determination (POD) of the satellite. The occultation antennas have the capability to track both GPS L1CA/L2C/L2P and GLONASS L1/L2 signals, providing a comprehensive set of data for atmospheric profiling, while the POD antenna is exclusively focused on tracking GPS satellites. The versatility of the Sentinel-6A satellite is further highlighted by its software-configurable receiver, which allows NASA/JPL for adjustments in tracking behavior to optimize data collection based on mission needs.

This paper offers a comprehensive evaluation of the EUMETSAT processed Sentinel-6A bending angle products using the Non-Time Critical Radio Occultation (RO-NTC) data from September 1[st] to December 31[st], 2021. NASA/JPL also provides bending angle profiles in Near-Real-Time (NRT) for which some discussions can be found in von Engeln (2024). During this





period the Sentinel-6A RO-NTC processor processed over 112,000 quality checked bending angle profiles. Approximately 62% of these occultations were from GPS signals, with the remaining being from GLONASS satellites. This distribution primarily reflects the varying sizes of the two satellite constellations, GPS with up to 32 satellites and GLONASS with nominally 24 satellites. However, several GLONASS satellites are not providing a second frequency and are thus unusable for RO (currently, affecting 3 satellites). Additionally, there is a slight imbalance between the number of setting and rising occultations, with setting occultations constituting about 53% of the total. This discrepancy can be attributed to two main reasons. The first reason is the different hardware configurations of the rising and setting occultation antennas, as detailed in Section 2. The second reason is related to the reboots of the RO instrument (some of them are reported in Section 2.1.1) which can affect the data collection and lead to variations in the number of occultations captured during different periods. Specifically, after an instrument reboot, GPS ephemeris data require more time to be downloaded by the zenith antenna, whereas GLONASS information is stored on-board. This long download time can affect the schedule of upcoming rising occultations.

Examining the daily count of nominally processed occultations, the mission's requirement of 770 bending angle profiles per day is exceeded with a 99.9% availability rate. This accomplishment underscores the satellite's operational reliability and its capability to provide comprehensive data coverage.

The analysis presented in this paper extends also to the Signal-To-Noise Ratio (SNR), where findings reveal exceedingly high SNRs coupled with remarkably low phase noise. This combination is crucial for generating bending angle products of high quality.

The EUMETSAT Sentinel-6 RO processor uses the Canonical Transform 2 (CT2) algorithm, as described by (Gorbunov and Lauritsen, 2004), to retrieve bending angles from the complex field measurements along the trajectory of the LEO satellite. CT2 is part of the radio-holographic inversion methods designed to handle multipath propagation problems of RO signals in the moist lower troposphere. As (Sokolovskiy et al., 2010) reported, inversion errors are influenced by the length of recorded RO signals and their noise levels. Implementing an effective L1/L2 signals cutoff strategy, based on the SNR, is crucial in the lower troposphere to enhance the reliability of the retrieved bending angle profiles.

Furthermore, the quality of bending angles is also affected by the cutoff and extrapolation of the L2 frequency, which is more susceptible to SNR reduction in the lower troposphere. The EUMETSAT Sentinel-6 RO processor follows to the ionospheric correction procedure described in (Culverwell and Healy, 2015), which includes an improved version of the L2 cutoff strategy. The various approaches for the L2 cutoff strategy implemented by the EUMETSAT Sentinel-6 RO processor are detailed in Section 2.1.4. The effectiveness of these strategies, including the robustly fitted ionospheric model used as a reference for determining the L2 signal cutoff point, is confirmed by comparing operational data with a reprocessed version.

Enhancements applied to the Sentinel-6A RO processor for effectively removing navigation bits from the received L1/L2 GNSS signals are discussed. This will be shown to have contributed to an overall improvement of bending angles data quality when data are compared against ECMWF short-range forecasts. In addition, interference signals on the I and Q components of the L2 frequency, not related to GNSS navigation bits, will be discussed trying to give a possible geographic localization of the interference sources.



The text is organized as follows: section 2 describes the satellite, with a focus on the RO experiment; section 2.1 provides a detailed description of the data used in this work, namely the EUMETSAT RO bending angle profiles stored in the Level 1B products, and the reference data used for their validation; section 2.1.1 is dedicated to the assessment of the daily occultation numbers and the impact of each GNSS satellite on the entire data set; section 2.1.2 focuses on the analysis on the SNRs recorded by the RO receiver for each constellation, transmitter satellite and tracked signal; section 2.1.4 reports about L2 signal cut-off strategy and extrapolation in troposphere; section 2.1.3 describes the biasing effect in troposphere related to the L1/L2 signal cut-off based on SNR; section 2.1.5 shows the impact of some improvements applied to the signal navigation bits removal algorithm on the bending angle statistics; section 2.1.6 talks about some some findings related to the presence of interference signals on the L2 frequency; section 3 presents an exhaustive validation of the bending angle profiles; section 4 provides the conclusions and a final outlook.

## 2 Satellite and instrument

The Sentinel-6A satellite flies in the so-called altimetry reference orbit, which was initially chosen for the TOPEX/Poseidon mission to have a good coverage of the oceans while at the same time avoiding aliasing effects among the tidal frequencies (Fu et al., 1994). It is a $66°$-inclined non-Sun-Synchronous orbit with an altitude between 1336 and 1356 km, which corresponds to a repeat cycle (i.e., the time it takes for the spacecraft to fly over the same patch of the surface) of 9.9 days (Donlon et al., 2021b, a). For the RO instrument, this translates to a local solar time coverage that does not present the clear pattern typical of sun-synchronous orbits. Figure 1 illustrates this point by combining RO events distributions for the Sentinel-6A and the EUMETSAT Polar System (EPS) Meteorological Operational (Metop) satellites, these latter flying on a $98.7°$-inclined Sun-Synchronous orbit at a mean altitude of 817 km. Note that the Sentinel-6A coverage changes over time while it is fixed for the Metop satellites. Furthermore, there are a number of linear features in the Sentinel-6A coverage, which are mostly GLONASS occultations, resulting from the similar inclination of the orbit planes occupied by the Russian constellation (about $65°$).

The Sentinel-6A satellite, showcased by a 1:1 scale model at the EUMETSAT entrance as depicted in figure 2, is equipped with a suite of instruments primarily designed to support the functionality of the Poseidon radar altimeter, its principal payload. Notably, the Radio Occultation (RO) experiment aboard S6A is considered a mission of opportunity, offering substantial benefits for numerical weather prediction (NWP) models and climate studies at a relatively minimal cost, as discussed in Harnisch et al. (2013) and Cardinali and Healy (2014).

The TriG GNSS-RO instrument, engineered by NASA/JPL and cited in works by Tien et al. (2010), Esterhuizen et al. (2009) and (peng Ho et al., 2020), tracks in Open-Loop mode a variety of signals: the traditional L1CA and L2 Codeless signals, the newer L2C signal from GPS, and the GLONASS Frequency Division Multiple Access (FDMA) signals on both L1 and L2 bands. The instrument's antenna configuration is optimized for data collection: the forward-looking antenna is arranged in a 3x2 array, while the aft-looking antenna has a 3x4 array, providing higher antenna gain for setting occultations, a factor contributing to the discrepancy in the number of setting versus rising occultations. Additionally, the zenith-looking antenna, a single-patch antenna, is dedicated to tracking GPS satellites for precise orbit determination. In a similar way, the COSMIC-2 mission also





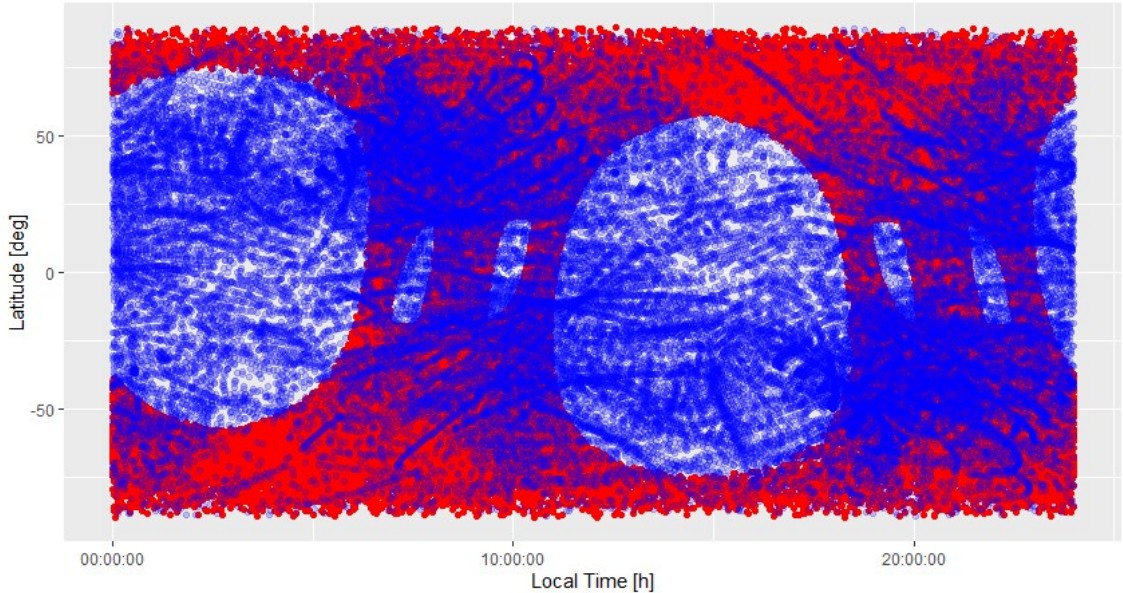

**Figure 1.** Local solar time coverage for Sentinel-6A (blue) and Metop-A/B/C (red) for October and November 2021.

utilizes the same receiver technology. Indeed, the COSMIC-2 satellites are equipped with the same TriG GNSS-RO instrument, having then the same tracking capabilities as Sentinel-6A.

The Sentinel-6A RO instrument's performance, is required to provide at least 770 quality checked profiles per day. The health of the GNSS constellations significantly affects the number and quality of RO profiles. While the GPS constellation maintains robust performance with more than 30 satellites, the GLONASS system has shown reduced reliability due to aging satellites and slow replenishment rates. To mitigate the impact of GLONASS degradation and enhance overall data collection, there are plans to enable the Sentinel-6A instrument to track Galileo L1/L5 signals, which will increase the quantity and quality of occultation data. The experience from the Sentinel-6A mission underscores the importance of having a multi-constellation tracking strategy. This is important in order to ensure the continuous and reliable acquisition of radio occultation data for weather forecasting and climate assessments.

## 2.1 Data and processing

The Sentinel-6A GNSS-RO dataset, utilized in this paper, spans from September $1^{st}$ to December $31^{st}$ 2021, and was sourced from the Sentinel-6A RO-NTC operational processor, which code is based on the Yet Another Radio Occultation Software (YAROS), a EUMETSAT developed and maintained RO processor. It employs the Canonical Transform 2 (CT2) (Gorbunov and Lauritsen, 2004) to deduce bending angles from the complex field measurements along the trajectory of the LEO satellite. As detailed in von Engeln (2022c), prior to calculating bending angles, the Sentinel-6 TriG receiver raw measurements are subjected to a preliminary data reconstruction step, made by using the JPL provided L0 decoder software. This provides





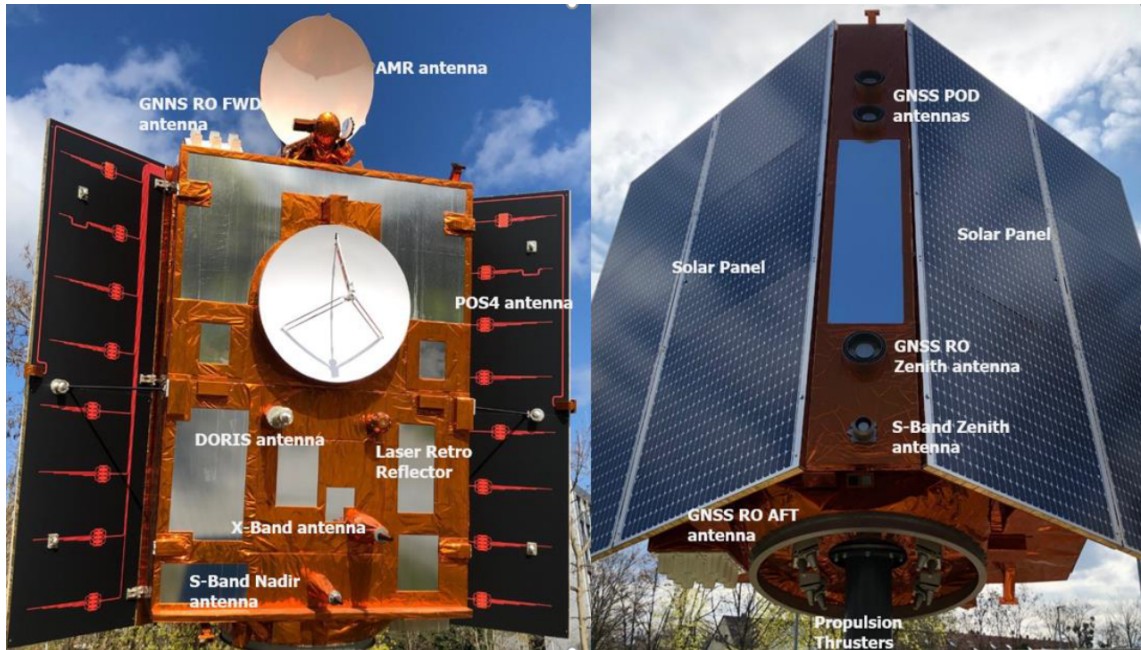

**Figure 2.** Picture of the 1:1 model of the Sentinel-6A satellite on display on the EUMETSAT campus (von Engeln, 2024). Left: nadir-looking side. Right: zenith-looking side. The main elements are labelled.

files containing occultation radio occultation data together with RINEX-3 files containing zenith antenna data. Subsequent reconstruction and calibration steps include signals SNR and phase reconstruction, Precise Orbit Determination (POD) using the Bernese GNSS Software v5.3 (Dach et al., 2015) developed by the Astronomical Institute of the University of Bern (AIUB),

measurement times and signals phase corrections to remove receiver and transmitter clock bias and removal of navigation bits by using NASA/JPL provided navigation bit files. Notably, navigation bit removal is mandatory for L1 in GPS and for both L1 and L2 in GLONASS signals. As outlined in section 2.1.5, the GPS L2P signal necessitates a distinct process to eliminate half cycle slips from the raw phase. Bending angle profiles, stored into level 1B products which format is specified in von Engeln (2022b), are available at the EUMETSAT website https://navigator.eumetsat.int/start.

To ensure the generation of high-quality and reliable POD solutions, NASA/JPL has activated the provision of essential auxiliary data. This includes GPS and GLONASS orbits and clock bias files, Earth Orientation Parameters (EOP) files, and LEO/GNSS attitude data files. The GPS and GLONASS orbital data are updated every 15 minutes, while GPS clock biases are provided at 30-second intervals. Notably, GLONASS clock biases are supplied at a more frequent 1-second rate, a critical factor for enhancing the quality of bending angle products derived from GLONASS occultations (Padovan et al., 2024). For a

detailed list of the auxiliary data files received by NASA/JPL and the ones generated by the Sentinel-6A RO-NTC processor, available on the EUMETSAT website, please consult von Engeln (2022a).





| Time | Description |
| --- | --- |
| 01/09/2021 05:01:30 | Yaw Flip Backward Configuration |
| 03/09/2021 05:07 | Data gap, duration 113 min |
| 05/09/2021 06:33:00 | Yaw Flip Forward Configuration |
| 09/09/2021 13:22 | Data gap, duration 45 min |
| 08/10/2021 09:41 | Data gap, duration 60 min |
| 19/10/2021 07:59 | Data gap, duration 52 min |
| 25/10/2021 08:18 | Data gap, duration 37 min |
| 05/11/2021 16:18 | Data gap, duration 34 min |
| 05/11/2021 16:24:28 | Yaw Flip Backward Configuration |
| 09/11/2021 17:28:38 | Yaw Flip Forward Configuration |
| 16/11/2021 11:00 | Data gap, duration 32 min |
| 16/11/2021 11:03:40 | GNSS-RO Script Update Start Period |
| 16/11/2021 11:23:01 | GNSS-RO Script Update End Period |
| 25/11/2021 16:48 | Data gap, duration 70 min |
| 26/12/2021 17:00 | Data gap, duration 63 min |
| 28/12/2021 19:15 | Data gap, duration 30 min |
| 30/12/2021 15:29 | Data gap, duration 35 min |

**Table 1.** GNSS-RO instrument activities performed between September and December 2021).

Throughout the period under study, routine operations and specific instrument activities were conducted, some of which influenced the daily count of produced occultations and the overall data quality. Table 1 details the instrument-related events and issues that impacted the RO data during this time frame, for the periods where occultations were not recorded for duration exceeding 30 minutes. For a more exhaustive description of instrument activities recorder since the beginning of the mission, refer to von Engeln (2024).

The validation process involved comparing the statistics of two operational RO missions: the 2 Metop B/C satellites with their GNSS Receiver for Atmospheric Sounding (GRAS) instruments, providing about 700 occultations per day each, and SPIRE, which operates a fleet of 3U cubesats (https://spire.com/spirepedia/cubesat/), equipped with the STRATOS GNSS RO instruments. SPIRE occultations are observed by a diverse fleet of satellites operating in both high inclination, near-polar orbits and low inclination, equatorial orbits. This strategic deployment improves data coverage in terms of latitude and longitude, as well as improving temporal coverage in terms of local time. These statistics were benchmarked against operational short-range forecasts from the European Centre for Medium-Range Weather Forecasts (ECMWF). For both datasets, bending angle profiles were thinned to 247 vertical levels. Metop-B and Metop-C occultations, available through the EUMETSAT ground segment, were analyzed for the specified period. In the case of SPIRE, four satellites were selected to generate statistics to ensure a comparable number of occultations with Metop and Sentinel-6.



The generation of statistics against ECMWF data involved co-locating profiles within a $0.5\,\mathrm{deg}$ grid resolution and applying forward modeling to bending angles using the RO Processing Package (ROPP). This forward-modeling process converts ECMWF data on temperature, humidity, pressure, and geopotential, provided at 137 vertical levels at the reference occultation

position, to 247 bending angle levels. In order to do that, ECMWF data are interpolated to the reference location in a bilinear fashion as described in (Consortium, 2021).

To demonstrate the enhancements in the Sentinel-6A RO-NTC processor software during the satellite commissioning phase, started after the satellite launch and concluded on November 19th, 2021, an additional validation was conducted using reprocessed bending angles from Sentinel-6A. This reprocessing used the latest Sentinel-6A RO-NTC processor version 4.0,

scheduled for operational deployment in the second quarter of 2024. Reprocessing was also applied to EPS and SPIRE data to maintain consistency across the datasets for comparison.

### 2.1.1 Daily occultations

Figure 3 displays the daily count of quality checked bending angle profiles processed by the operational EUMETSAT Sentinel-6A RO-NTC processor during the analyzed period, categorized by GNSS system. The black bars represent the combined total

of daily GPS and GLONASS occultations, and the black horizontal line marks the mission's target of 770 bending angle profile per day. To fully understand this figure, it should be examined alongside Table 1, which enumerates the activities carried out on the S6A satellite and the RO instrument. These activities have had an impact on the RO processing, and their inclusion in the analysis helps to contextualize the fluctuation in the number of daily occultations, providing a comprehensive view of operational performance and any external factor affecting it.

Due to the three GLONASS orbit planes, the instrument exhibits fluctuations attributable to the changing visibility of satellites from the Low Earth Orbit (LEO) plane. These variations are a result of the limited number of GLONASS orbit planes, which affects the satellite visibility and, consequently, the number of occultations recorded. Conversely, the six GPS orbit planes provide a more stable and consistent satellite visibility from the LEO perspective, leading to less variation in the daily count of GPS occultations. This stability inherent in the GPS system ensures a more uniform distribution of occultations over

time.

The first significant decrease in the number of daily processed occultations was observed on September 2nd and 3rd, 2021. During this period, the RO-NTC processor was unable to process GLONASS occultations because of missing data in the GNSS orbit files provided by JPL. This lack of data led to gaps in the processing capability of the RO-NTC system, specifically affecting the handling of GLONASS signals and resulting in a noticeable reduction in the total number of occultations recorded

during these days. Subsequent Sentinel-6A RO processor updates made it more robust to these kind of issues.

On December 23rd, 2021, another significant reduction in daily occultations was recorded when the GNSS-RO instrument stopped tracking occultations from the GLONASS constellation with PRNs greater than 06. This issue led to a decrease of approximately $50\%$ in the total number of GLONASS occultations, persisting until January 2nd, when the instrument was reset. Although not depicted in Figure 3, post-reset, the instrument functioned normally until January 8th, when the problem

recurred. A subsequent instrument reboot temporarily addressed the issue, but it was not until a series of software updates





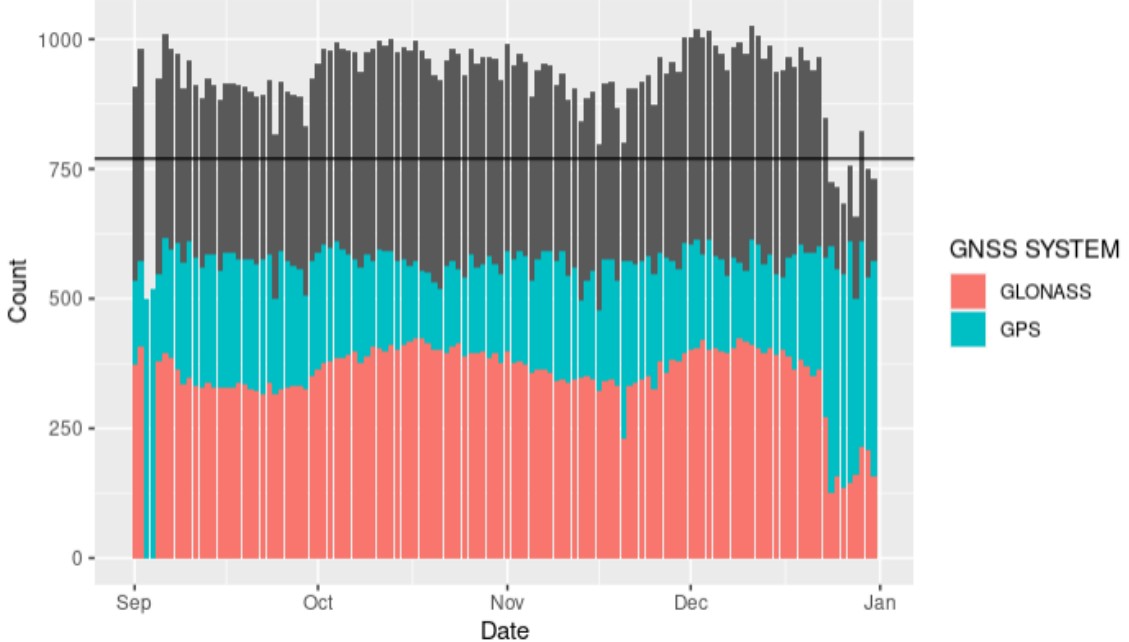

**Figure 3.** Daily number of quality checked bending angles profiles per constellation in the analyzed period. The black bars indicate the sum of daily GPS and GLONASS occultation profiles, while the black horizontal line indicates the mission requirement of 770 occ/day.

between January 12th and 13th, 2022, that the problem was definitively resolved. Further investigations into the cause of the issue revealed that the instrument's malfunction was due to the use of stored, outdated, and no longer updated GLONASS ephemerides, which impacted its proper operation.

Minor decreases in daily occultations were noted during updates to the instrument's configuration for handling yaw flips.
During these maneuvers, the satellite is rotated 180 degrees, causing the antenna that normally tracks setting occultations to track rising occultations, and vice versa. These yaw flip phases, which typically last less than 10 minutes, can lead to degraded or unavailable occultations. The brief duration of these flips means that their impact on the total daily occultations is limited, but they can still cause noticeable fluctuations in the data during the periods when the satellite is reorienting itself.

Minor reductions in daily occultations can also result from data gaps in the raw level 0 data provided by the RO instrument.
These gaps may arise from various causes, including autonomous or commanded instrument reboots, often triggered by the loss of GLONASS tracking, updates to the instrument's software, downtimes in the data link, or any other incidents leading to the loss of level 0 data. Table 1 details these data gaps, specifically highlighting instances where no occultations were recorded for durations exceeding 30 minutes. These interruptions in data acquisition contribute to fluctuations in the number of daily recorded occultations and are crucial for understanding the operational stability and reliability of the RO instrument.

Since the start of the mission, the average number of data gaps has been approximately 0.13 per day, leading to a total data loss of around 1.5%. This figure encompasses the period of early instrument activation, during which data gaps were more





frequent. However, focusing on the period from June $10^{th}$, 2021, onwards, which represents more stable, nominal operations, the situation improves. From this date, the average number of data gaps decreases to about 0.11 per day, with the total data loss reduced to approximately $0.5\%$. This indicates an enhancement in the reliability and efficiency of data collection as the mission progressed beyond its initial activation phase.

### 2.1.2 SNR analysis

The analysis of the SNR of the GNSS signals tracked by the RO instrument is an important aspect of radio occultation retrievals. As discussed in the works of Gorbunov et al. (2022a) and Gorbunov et al. (2022b), higher SNRs improve the tropospheric penetration or RO profiles. The effect of SNR to the random error when RO profiles are compared against Numerical Weather Prediction data is mission dependent, showing some saturation features larger for some missions and smaller for others. It´s worth nothing that lower SNR missions like SPIRE have demonstrated the ability to systematically detect key atmospheric features. In this section, we aim to delineate the SNR characteristics specific to the Sentinel-6 RO receiver, contributing with its good performance to the field of atmospheric observations.

The histograms in Figure 4 display the distribution of Sentinel-6A mean SNRs, highlighting that GLONASS signals exhibit higher SNRs compared to GPS signals, attributed to the not required code-less tracking for GLONASS. To fully understand the SNR distributions, Figure 4 should be analyzed in conjunction with Figures 5 and 6. Figure 5 presents the mean SNRs plotted per PRN, averaged over the analyzed period, and categorized by constellation and frequency. Figure 6 shows the daily averages of mean SNRs, grouped according to the tracked GNSS signal code. The GPS distributions show lower SNRs compared to GLONASS, more visible for the L2 plot. The different gains of the rising and setting antennas are slightly more pronounced for L1 plot than for L2 plot, peaking at about 1250 V/V and about 1000 V/V, as also noticeable in the GPS-related panels of Figure 6.

In contrast, the SNR distributions in Figure 4 for GLONASS are tri-modal for both L1 and L2 frequencies. Notably, peaks in L1 around 500 V/V are influenced by satellites R13, R19, R20, and R22, as seen in the bottom left panel of Figure 5. Additionally, for L1, there are peaks at approximately 1000 V/V and 1700 V/V, while for L2, the peaks are at around 300 V/V, 750 V/V, and 1250 V/V, influenced by satellites R13, R16, and R22 as seen in the bottom right panel of Figure 5.

The analysis of GPS signals per PRN, as shown in Figure 5, indicates that while some PRNs perform better than others, the SNRs and tracking capabilities of GPS signals remain relatively stable across different satellites. This consistency applies to both L1 and L2 GPS signals, with no significant distinction observed between the tracking of L2P and L2C signals, a finding confirmed by the top right and bottom left panels of Figure 5.

In contrast, GLONASS exhibits more variability in SNR values among its satellites. Specifically, L1 is tracked with an SNR lower than 500 V/V for satellites R13, R19, R20, and R22, and L2 shows similar SNR values for satellites R13, R16, and R22. Notably, for satellites R13, R19, and R20, the SNRs are more favorable for L2 than for L1. This variation in GLONASS SNRs indicates a disparity in signal strength and tracking efficiency across its constellation, contrasting with the more uniform performance observed in the GPS system.



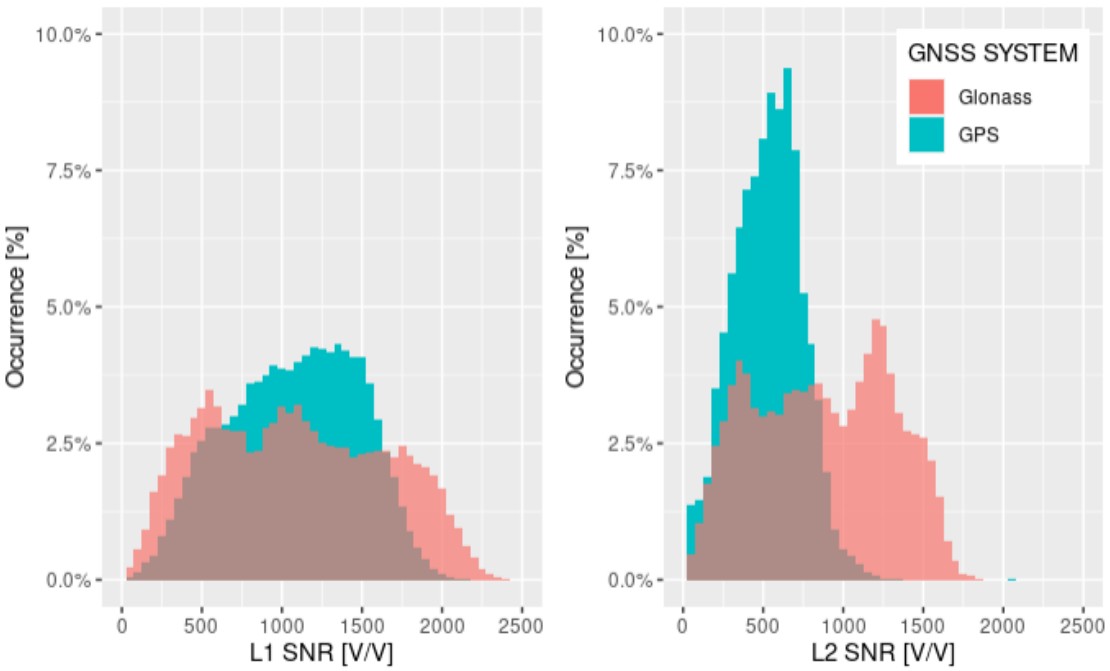

**Figure 4.** Histograms of mean SNRs measured between 60 and 80 km SLTA split per constellation, for L1 (left) and L2 (right) frequencies.

The further examination of mean observed SNRs involves daily average calculations, segmented by tracked signal and differentiated between setting and rising occultations. These results are depicted in Figure 6. The time series in this figure prominently illustrates the impact of yaw flips executed in early September and November, which resulted in diminished SNR, being the yaw flip not followed by the beam forming adjustment. It´s worth nothing, comparing Figure 4 with Figure 6 that, during yaw flips the tracking capabilities remained unchanged. This is confirmed by the yaw flip event that occurred between

November $5^{th}$ and $9^{th}$ 2021. The yaw flip in early September also demonstrates the robust tracking abilities, although one has to exclude September $2^{nd}$ and $3^{rd}$. On these days, the absence of GNSS orbit files from JPL resulted in the processor's inability to process GLONASS occultations asreported in Section 2.1.1.

Moreover, Figure 6 highlights the variability in tracking capabilities for GLONASS signals over time, which is influenced by the geometrical configuration of the orbits. This variability contrasts with the more stable tracking performance of GPS signals,

underscoring the sensitivity of GLONASS tracking to its orbital geometry. Such temporal fluctuations in SNR for GLONASS signals emphasize the dynamic nature of satellite tracking performance and the importance of considering orbital geometries in the analysis of GNSS data quality.

To provide a thorough assessment of the Sentinel-6A TRIG RO instrument, it is essential to examine not only the SNR values but also the phase noise impacting the tracking of GNSS signals (Withers, 2010). Phase noise represents the random

variations in the phase of the tracked signals, arising from different sources, including electronic components or imperfections





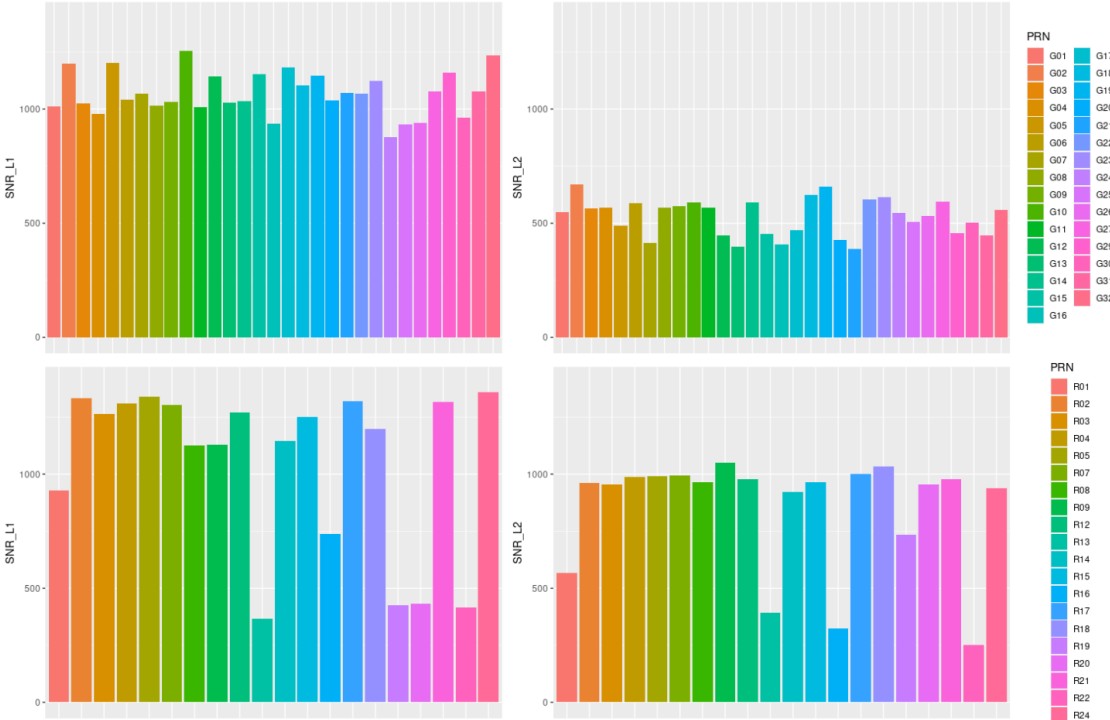

**Figure 5.** Mean SNRs measured between 60 and 80 km SLTA plotted per PRN, averaged over the analysed period, split for GPS (top) and GLONASS (bottom) and for L1 (left) and L2 (right) frequencies.

in the receiver. This noise can distort the received signal, leading to inaccuracies in the estimation of phase delays induced by the atmosphere.

High phase noise can negatively affect the SNR by adding random fluctuations to the received signal. Conversely, a low SNR can make the tracking more vulnerable to phase noise, highlighting the interdependent nature of these two parameters. Major

contributors to the bending angle error budget in the upper stratosphere include phase noise as well as the the Precise Orbit Determination (POD) solution (Padovan et al., 2024) and residual ionospheric errors (Gorbunov, 2002; Danzer et al., 2013, 2015; Healy and Culverwell, 2015) remaining after the ionospheric contribution correction. Figure 7 shows the histograms of the excess phase noise distribution for L1 and L2, calculated per occultation between 60 and 80 km SLTA. This figure reveals that the RO instrument experiences lower phase noise levels when tracking GLONASS occultations compared to GPS ones.

In comparison to the other two missions discussed in this paper, Sentinel-6A exhibits phase noise levels that are slightly higher than those of the GRAS instrument, with an average around 0.08 mm. However, these levels are significantly lower than those of SPIRE, where the phase noise typically ranges between 0.5 mm and 1.3 mm. This comparison highlights the relative performance of Sentinel-6A in terms of phase noise, situating it between the lower noise levels of GRAS and the higher levels associated with SPIRE. The data suggests that Sentinel-6A maintains a good balance in phase noise performance,





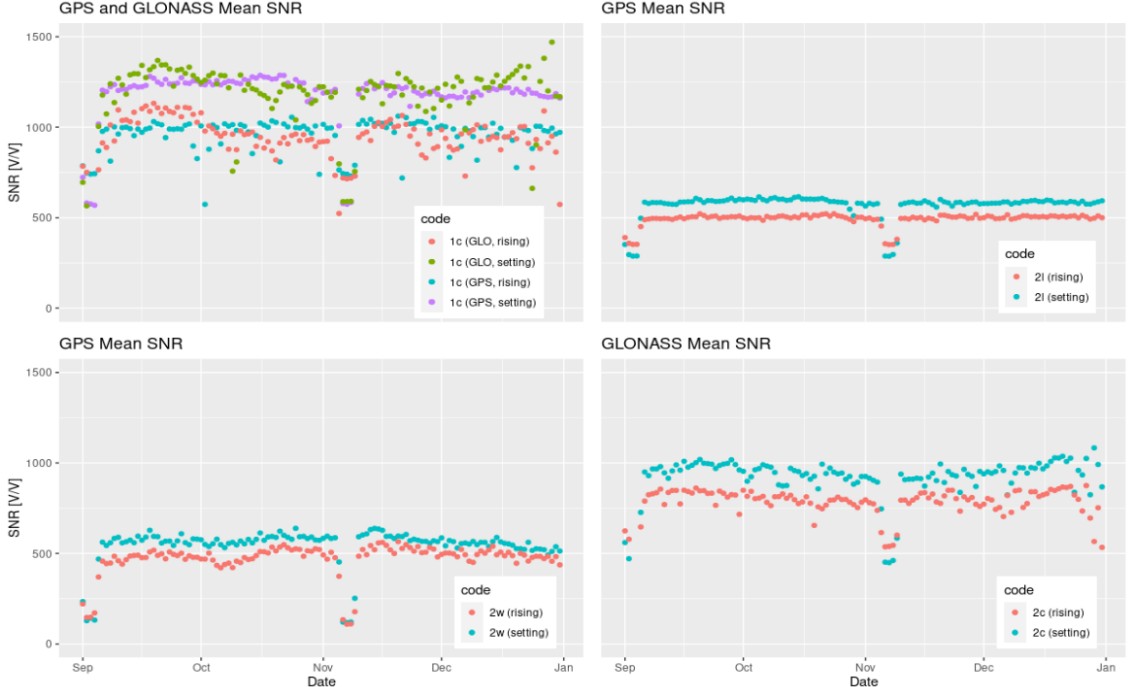

**Figure 6.** Daily averages of mean SNRs measured between 60 and 80 km SLTA split for tracked GNSS signal.

offering a more favorable operational condition for atmospheric sounding compared to SPIRE, while slightly lagging behind the performance of GRAS.

### 2.1.3 L1/L2 signals cut-off based on SNR

Over the past decade, the tracking capabilities of GNSS receivers used for positioning purposes have significantly improved, and RO GNSS receivers are not excluded (Gill et al., 2023). In particular, the Open-Loop tracking mode represents a major evolution of the tracking function (Mohamady and Amiri, 2013; Tien et al., 2010; Ao et al., 2009). This mode leverages on a priori knowledge (model) of the atmospheric delay, or equivalently, the atmospheric Doppler, which the receiver uses to initially estimate the received GNSS signal frequencies.

The introduction of Open-Loop tracking in RO receivers has notably enhanced their tracking capabilities, especially in the moist lower troposphere, where Closed-loop tracking mode encounters problem related to the Phase-Locked Loop (PLL) tracking technique. Open-Loop tracking mode facilitates improved tropospheric penetration (Sokolovskiy et al., 2009), enabling more reliable and accurate data collection in these regions. Currently, Open-Loop tracking mode is widely employed in many RO missions, sometimes extending to the middle and upper stratosphere, or even completely replacing the Closed-Loop mode as it is for Sentinel-6 RO receiver.





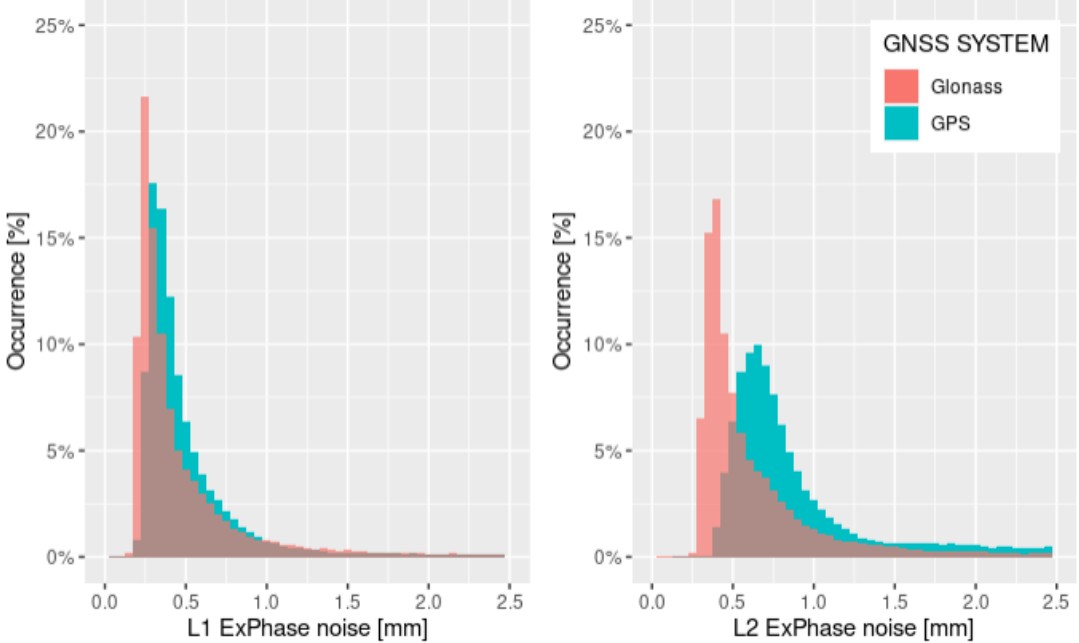

**Figure 7.** Histograms of Excess Phase noise measured between 60 and 80 km SLTA split per constellation, for L1 (left) and L2 (right) frequencies.

In Open-Loop tracking mode, an important characteristic of the GNSS signals is the presence of long tails at the lowest
Straight Line Tangent Altitudes (SLTA), which are sometimes primarily composed of noise. This is the case of Sentinel-6A RO instrument, set to reach SLTA values down to -350 km. Despite the EUMETSAT RO processor employs the fast phase transform technique to minimize inversion errors in the retrieval of bending angles, inaccuracies may still arise due to the length of the recorded signals and the associated noise. The cut-off height for the L1 and L2 signals can significantly influence the bias structure in the troposphere when occultation profiles are analyzed against reference data, such as ECMWF data
(Sokolovskiy et al., 2010). Therefore, the determination of the appropriate cut-off height becomes critical in ensuring the accuracy and reliability of tropospheric data obtained from GNSS signals in Open-Loop tracking mode.

     Modifying the cutoff points in the RO processing for L1 and L2 signals recorded in Open-Loop mode significantly affects the retrieval of bending angles in two ways. Firstly, the RO geometry imposes that rays with larger bending angles will be received later by the receiver, which in turns means that they will be received at lower altitudes. If the signal cutoff is set too high,
these late-arriving rays at lower altitudes will be excluded from the signal processing, potentially leading to an underestimation of bending angles at these heights. This could result in a more negative bias in the tropospheric bending angle profile when compared with ECMWF data, an effect that is particularly pronounced in lower latitude bands where the troposphere contains more moisture.





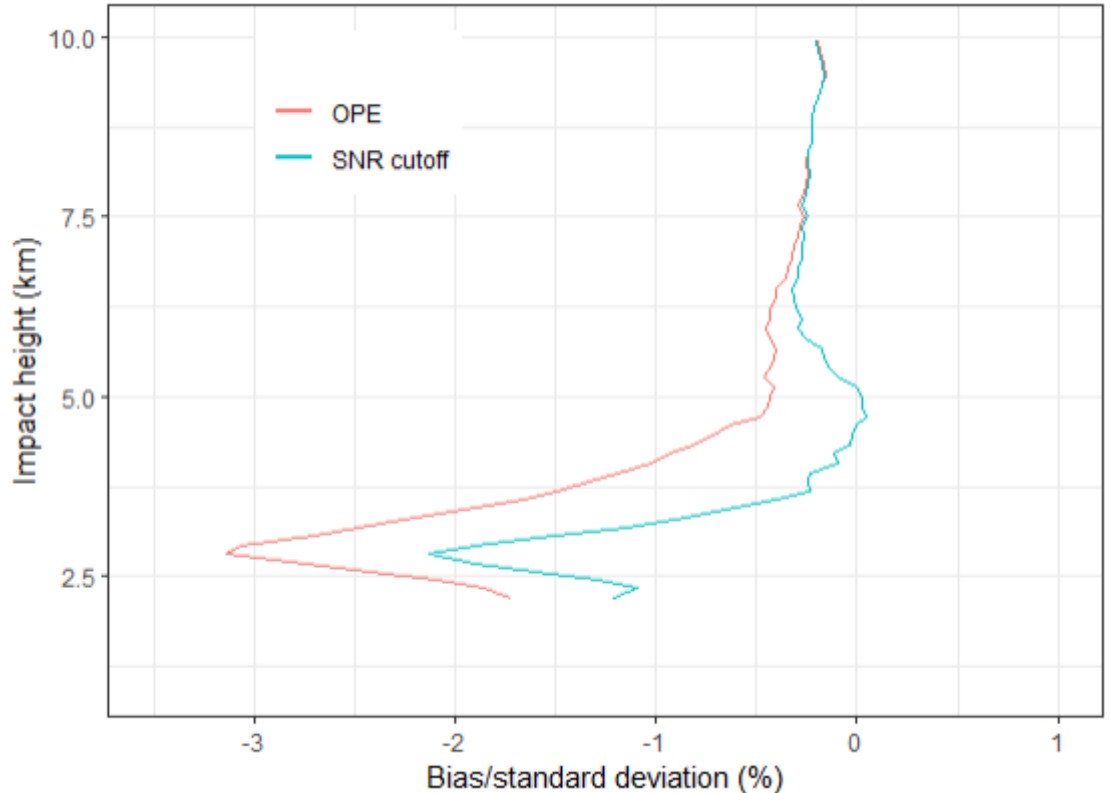

**Figure 8.** Global robust statistics of S6 bending angles compared against ECMWF short range forecasts during the analyzed period. Only vertical profiles of biases (or systematic deviations) in troposphere are showed for operational and reprocessed data employing the SNR-based cutoff algorithm.

Secondly, lowering the signal cutoff point too much might allow the inclusion of mostly noise (noise tail) in the bending angle processing, introducing a positive bias. Thus, finding the optimal cutoff point is a delicate balance: it must be high enough to exclude noise but low enough to capture the complete signal, especially in regions with high moisture content.

The Sentinel-6 RO processor employs the SNR-based signal cutoff algorithm for L1 and L2 signals, as detailed in Sokolovskiy et al. (2010), with minor modifications to address the zero amplitude/SNR occurrences regularly noted in L2 data from JPL receivers. Figure 8 demonstrates the impact of this implementation on the bias structure within the tropospheric heights, revealing a shift of about $1\%$ to the right when the data are compared against ECMWF.

### 2.1.4 L2 signals cutoff and L2 extrapolation in troposphere

After the launch of the Sentinel-6A satellite, during its commissioning phase and subsequently, investigations were initiated to address issues related to the cut-off of L1/L2 GNSS signals and the extrapolation of the L2 signal performed by the Sentinel-6A RO processor in the troposphere. These investigations were prompted by observations of excessively biased L2 frequency





data in the tropospheric segment of some occultations. The Sentinel-6A RO processor follows the ionospheric correction procedure outlined in (Culverwell and Healy, 2015). Given that the L2 signal is more susceptible to SNR reductions, this procedure suggests to cutoff the L2 signal in troposphere, where it becomes noisier, and extrapolate it to lower altitudes using an ionospheric model based on a Chapman Layer. This is done before the L1 and L2 bending angle profiles are merged to eliminate the ionospheric influence. The smooth transition of actual measurements with the extrapolated ionospheric model

occurs within a transition range, commencing from the L2 cutoff point and progressively blending into the real measurements. The determination of the L2 cutoff point was a significant focus of this investigation, aiming to enhance the accuracy and reliability of the RO tropospheric data processed by the Sentinel-6A RO processor.

The initial approach for processing the Sentinel-6A RO data involved using a fixed threshold of $50\mu$rad to determine the cutoff point for the L2 signal based on the L1-L2 bending angles difference. This method significantly improved the bending

angles processing, as indicated in Figure 9, showing the effect of enabling and disabling the L2 signal cutoff algorithm. However, it encountered issues during periods of high solar activity, where increased variability and a negative bias in the L1-L2 bending angle residuals often resulted in the L2 signal being truncated prematurely at too high an altitude. To counter this, a conservative height threshold of 30km was set to prevent the L2 signal from being cut above this altitude.

Figure 10 presents the density distribution of the L1-L2 difference for two distinct dates, December 10$^{th}$, 2021 (a period of

low solar activity) and December 23$^{rd}$, 2021 (a period of high solar activity), at various altitudes. Under the $50\mu$rad threshold with a maximum cut-off height of 30km, the L1-L2/L2 signals are typically truncated above 5km during low solar activity periods, with the cut-off occurring slightly higher during periods of high solar activity.

As observed and previously mentioned, employing a fixed threshold for the L2 signal cut-off, as depicted in Figure 10, can result in prematurely cutting the L2 signal, especially under varying solar activity conditions. To address this, an enhanced

implementation, which is now operational, adopts a robust mean of the L1-L2 bending angles as the reference point for applying the $50\mu$rad threshold to determine the L2 cut-off. This adjustment is made before the ionospheric correction process, enabling a more dynamic response to fluctuations in the L1-L2 bias induced by different levels of solar activity. This method allows for a more accurate representation of the ionospheric impact on the L2 signal, ensuring that the cut-off is applied more appropriately across varying solar conditions.

An alternative approach consists of integrating the L2 cutoff into the bending angles ionospheric correction process. This involves using a robustly fitted ionospheric model to the current data as the reference for determining the L2 signal cutoff point, in line with the suggestions in Culverwell and Healy (2015). Figure 11 presents normal statistical comparisons of bending angles against ECMWF data, showing the performance of a normal (or non-robust) and a robust ionospheric bending angle linear fitting algorithm. The data indicates that using the robust linear fitting algorithm can enhance the standard deviation

by up to 2.5% compared to operational data using the robust mean approach, above approximately 18km altitude. Between approximately 22km and 36km, the improvement in using robust fitting over non-robust fitting is modest but still significant to demonstrate the decreased sensitivity to strong ionospheric variations.

This investigation suggests that a robust linear fitting is better suited for handling irregular L1-L2 bending angles and is less impacted by noisy L2 data at lower tropospheric altitudes. Consequently, despite the modest improvement in the ran-





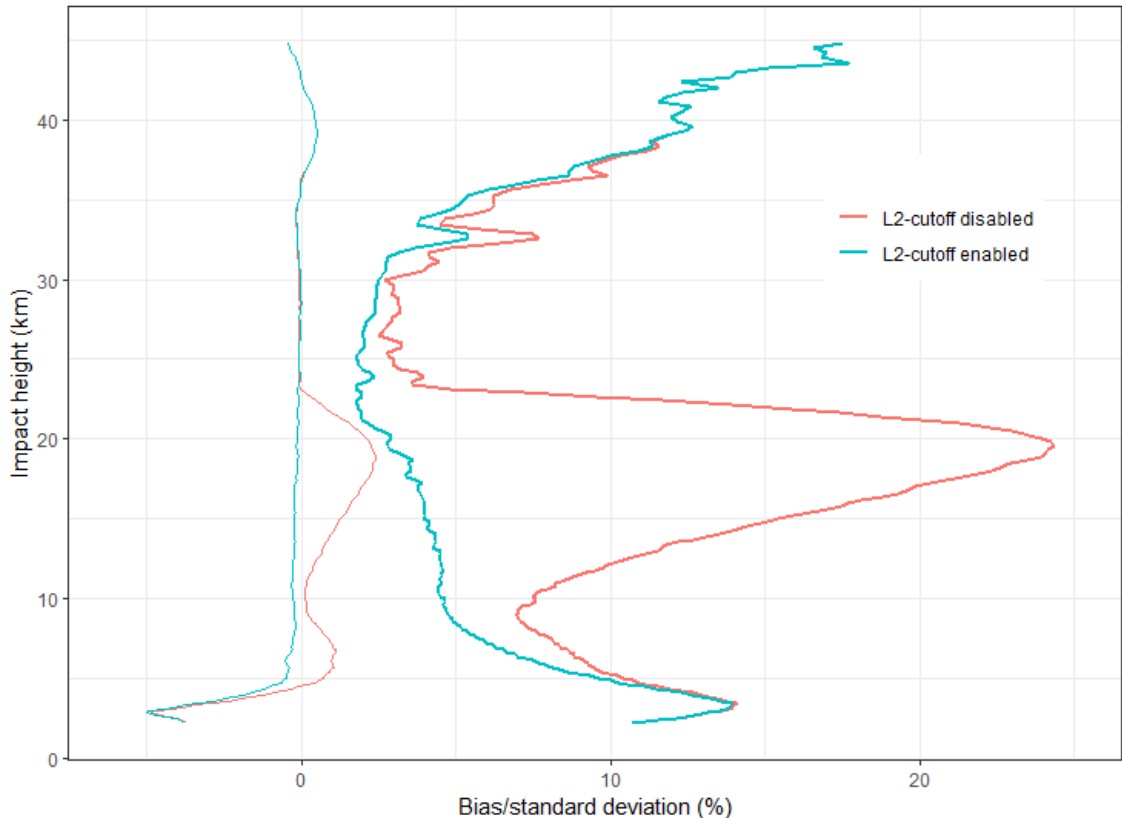

**Figure 9.** Global normal statistics of S6 bending angles compared against ECMWF short range forecasts during December 2021. Vertical profiles of biases (or systematic deviations) with respective standard (or random) deviation are showed for two runs: with (red) and without (blue) the L2 signal cutoff algorithm activated.

dom error, the robust fitting approach is still favored in presence of strong ionospheric disturbances, making it a worthwhile implementation choice for L2 signal cutoffs in the ionospheric correction phase.

### 2.1.5 Navigation bits removal

One of the preparatory steps before starting the bending angle retrieval algorithm involves removing navigation bits from the received L1/L2 GNSS signals. In the Sentinel-6 RO-NTC processor, this process makes use of the JPL-provided navigation 365 bits data stream. In the initial version of the Sentinel-6 RO processor, the navigation bits removal using JPL data was followed by an internal half cycle slips detection algorithm applied to all tracked GNSS signals, following the discriminator formulation suggested by Sokolovskiy et al. (2009).

An improved version of the navigation bit removal algorithm adopted by the Sentinel-6 RO processor makes use of the half cycle slips detection algorithm only for the GPS L2P signal, relying on the JPL provide navigation bit data file for the



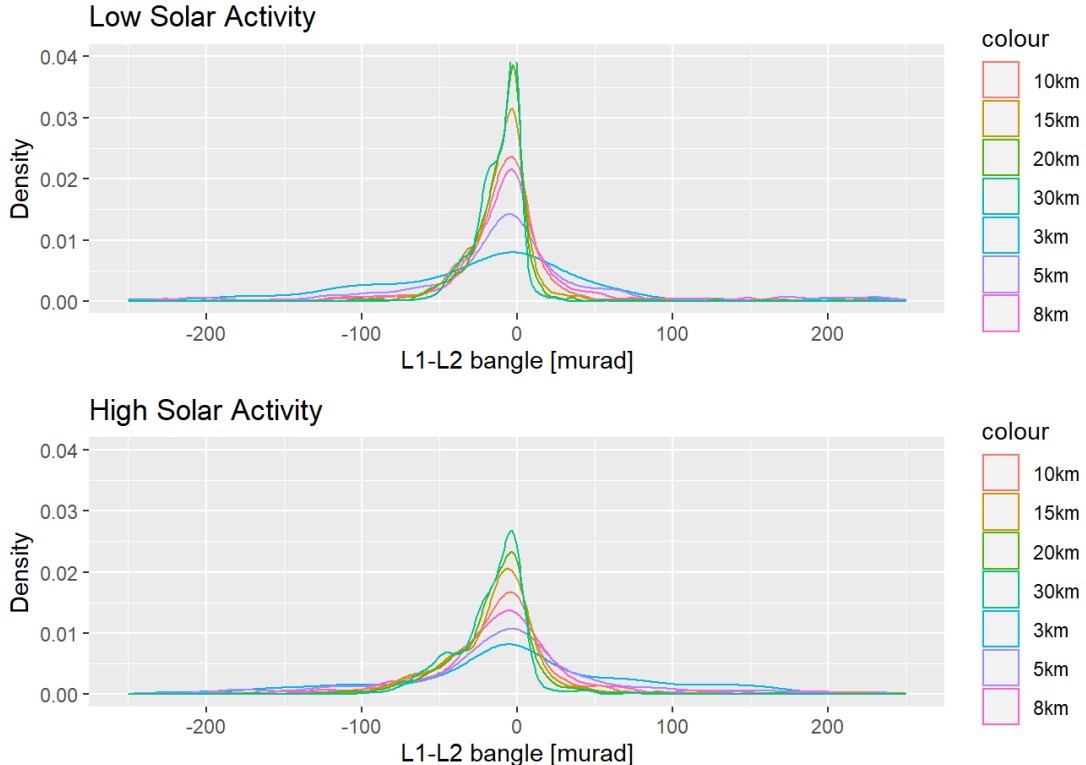

**Figure 10.** L1-L2 difference density distribution for December 10th 2021 (low solar activity) and December 23rd 2021 (high solar activity) at different heights.

other tracked signals. Figure 12 illustrates the impact of these improved algorithm, displaying the difference in a sample bending angles profile processed by using the revised algorithm and the original version. Noteworthy is the difference between bending angles in the lower troposphere, where lower SNR signals are tracked and the effects of multi-path processes are most pronounced.

Figure 13 shows the effect to the global bending angle normal statistics resulting from the refinement of the navigation bit removal algorithm, as data are compared against ECMWF models. Improvements to the random error extend from the troposphere between 5k and 23km impact height and to stratosphere above 30km impact height. These modifications have led to an overall enhancement in the quality of the generated bending angle profiles.

### 2.1.6   Interference on L2 signal

During the commissioning phase of the Sentinel-6A satellite and the routine quality assessment of its RO data, certain occultations revealed the presence of interference signals superimposed on the I and Q components, which were not associated with GNSS navigation bits. Specifically, interference signal at 40/20Hz were detected superimposed on the L2 signals, noticeable





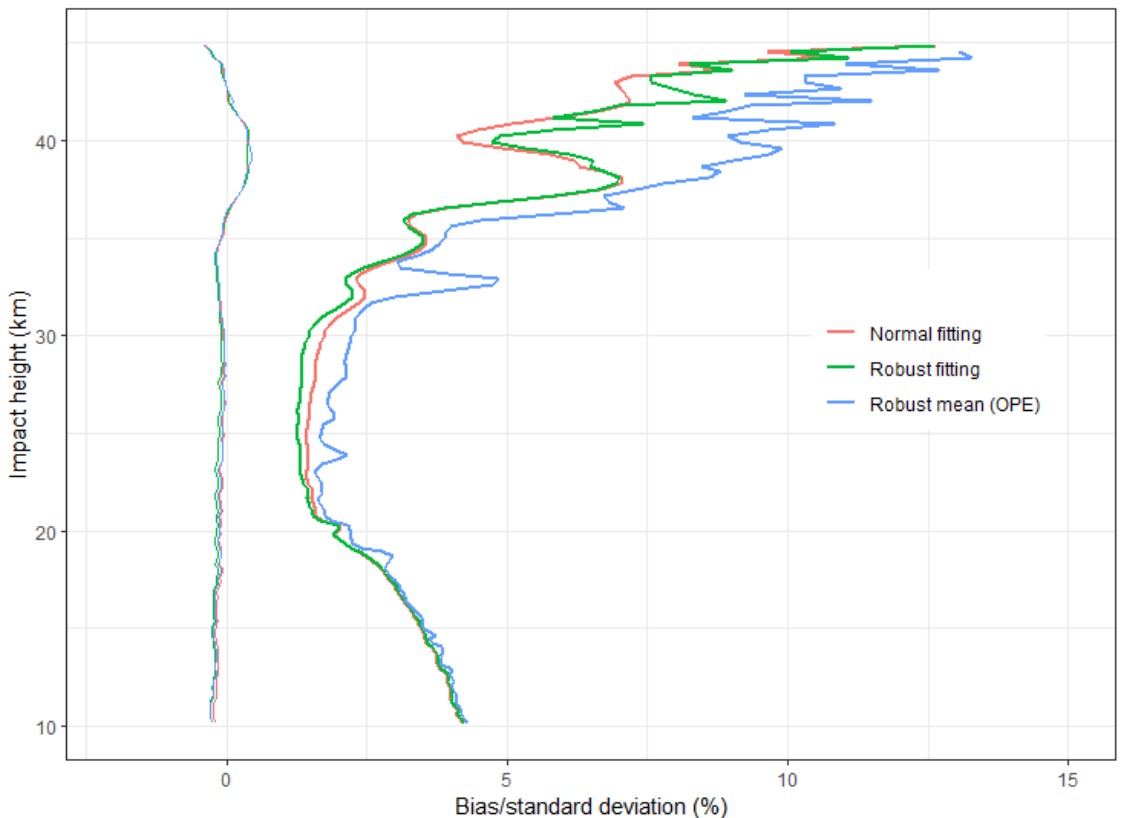

**Figure 11.** Global normal statistics of S6 bending angles compared against ECMWF short range forecasts during December 2021. Vertical profiles of biases (or systematic deviations) with respective standard (or random) deviation are showed for different choices of ionospheric model fitting algorithms.

in the SNR spectrum of these signals. Figure 14 presents the SNR spectrum for both L1 and L2 during a single occultation, highlighting these anomalous frequencies on the L2 plot.

Investigating the origin of interference observed in the SNR spectrum of the L2 signals is an interesting point for assessing
its impact on the quality of RO products. Understanding whether these spurious signals are geographically localized is equally important. Figure15 shows the RO receiver's geographical positions for all occultations recorded between August 2021 and May 2022, clearly indicating that these interference are more frequent when the Low Earth Orbit (LEO) satellite passes over the boundary between Russia and America. This observation is in line with findings from Isoz et al. (2014), which examined how terrestrial interference sources influence space-based GNSS receivers, with a specific focus on the GRAS instrument aboard
the Metop-A satellite. The study identified that terrestrial interference could induce pulsed interference and background noise fluctuations, without significantly compromising the GRAS data quality. Sentinel-6 data confirm these findings, most likely because the interference does not have enough power to affect the performance of the receiver significantly.





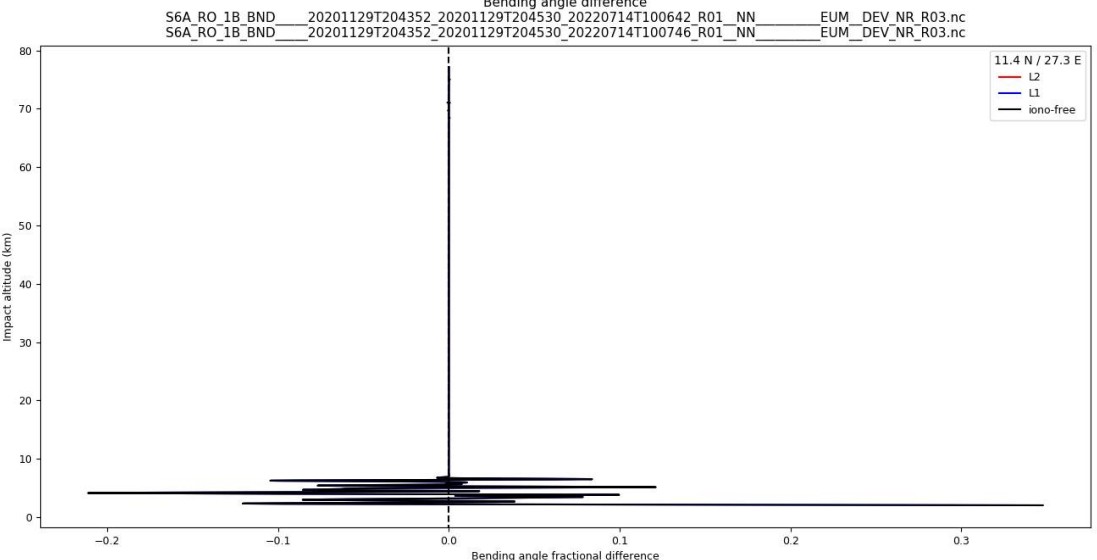

**Figure 12.** Difference between bending angles of a single occultation profile, retrieved by using the refined version of the navigation bits removal algorithm and the currently operational one.

## 3 Bending angles validation

Figure 16 presents the robust statistics of (O–B)/B, where O (Observable) represents the operational measurements from
Sentinel-6A, Spire, and GRAS B/C, and B (Background) is derived from the forward-propagated ECMWF short-range forecasts as a function of impact height. The deviations are expressed in percentage, facilitating a direct comparison of standard deviation values against the magnitudes of actual data at each height level. The data are analyzed using a robust estimator, as recommended by Hoaglin et al. (2000), which effectively mitigates the influence of outliers in noisy distributions, yielding standard deviation and the percentage of data points within the $\pm 2\sigma$ interval.

The analysis of Figure 16 clearly highlights the high quality of the operational Sentinel-6 bending angles, showing they are largely on par with those from the two EPS missions concerning both systematic and random errors. The standard deviation for Spire's occultations above 30 km is notably higher than that of the other missions, indicating greater data variability from Spire's RO receivers. While some of this discrepancy in the upper stratosphere may be attributed to the POD solution or residual ionospheric errors, the primary cause is the higher signal phase noise levels of Spire measurements, significantly influencing
the bending angle error budget at these heights. This finding is supported by the fact that the EUMETSAT RO processor uniformly smooths the bending angles for all missions, hence the inherent phase noise levels in the missions directly impact the bending angles profiles. The difference in the standard deviation between GRAS or Sentinel-6A and Spire data diminishes towards lower altitudes where other error contribution, in particular horizontal inhomogeneities, become the dominant drivers of RO uncertainty.



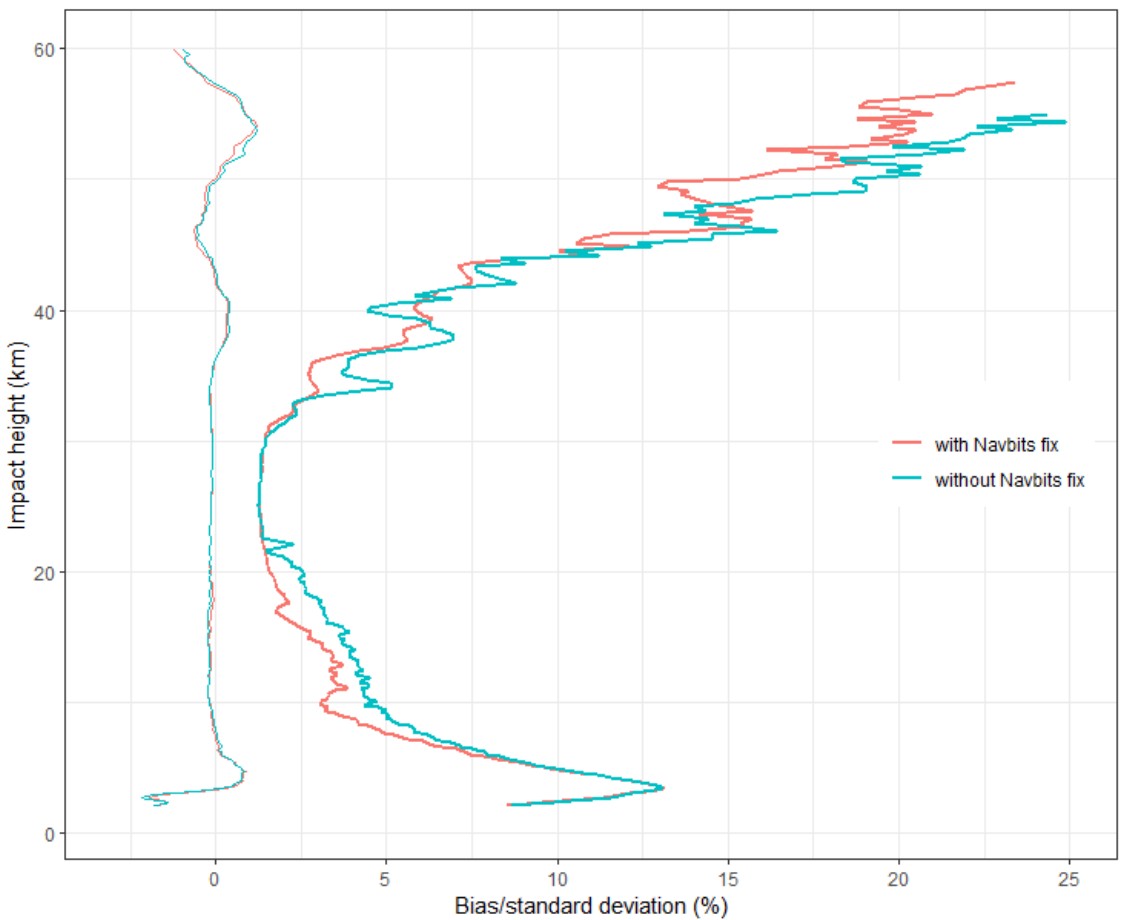

**Figure 13.** Global normal statistics of Sentinel-6A bending angles compared against ECMWF short range forecasts during the analyzed period. Vertical profiles of biases (or systematic deviations) with respective standard (or random) deviation are showed for different versions of the RO-NTC processor, with and without the navigation bite removal algorithm refinement. Statistics include both GPS and GLONASS occultations.

Concerning the vertical bias structure, all three missions show high consistency above approximately 7 km impact height (or about 5 km above sea level). Differences emerge in the troposphere, where the distinct instruments and their tracking conditions/modes, alongside the varied signal cut-off strategies, play an important role. Sentinel-6A exhibits a marginally larger negative bias compared to the two EPS missions, yet it is significantly less than that of Spire.

The operational RO data from Sentinel-6A, Spire, and GRAS B/C were reprocessed using the latest version of the EU-415  METSAT RO processor, set for deployment at the EUMETSAT Sentinel-6A RO-NTC facility in the second quarter of 2024. Figure 17 presents robust statistics from this reprocessing. Slight improvements in the standard deviation for Sentinel-6A are observed around 18km and above 50km, with the data now closely aligning with the GRAS measurements. The enhancement at 18km primarily results from the refined transition between the ionospheric model and actual bending angles measurements,

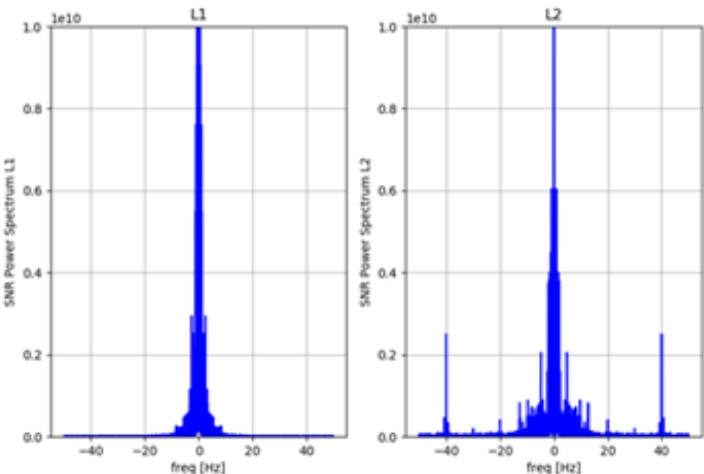

**Figure 14.** L1 (left) and L2 (right) SNR spectrum for a single Sentinel-6A GPS occultation recorder on October $1^{st}$, 2021.

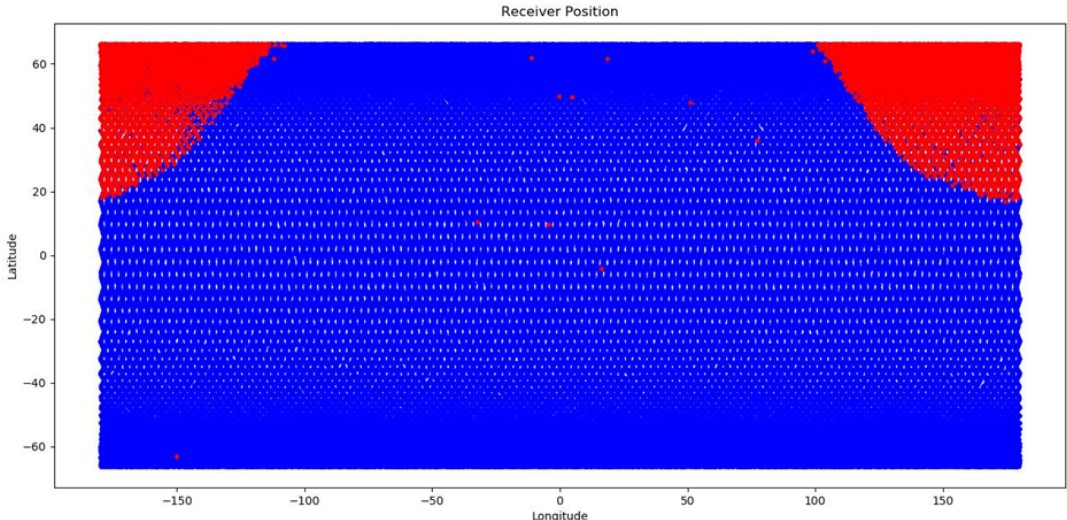

**Figure 15.** Receiver position for Sentinel-6A occultations affected by the interference signals in the L2 frequency, in the period between August 2021 and May 2022.

facilitated by the updated L2 extrapolation algorithm in the troposphere (discussed in section 2.1.4). The improvement above
50km comes from the correction made to the Sentinel-6A navigation bits removal algorithm (detailed in section 2.1.5).

An alternative way for inspecting the impact of the navigation bits removal algorithm correction in Sentinel-6A can be examining the bending angles statistics segregated by GNSS system, as depicted in Figure 18. This view highlights a reduction



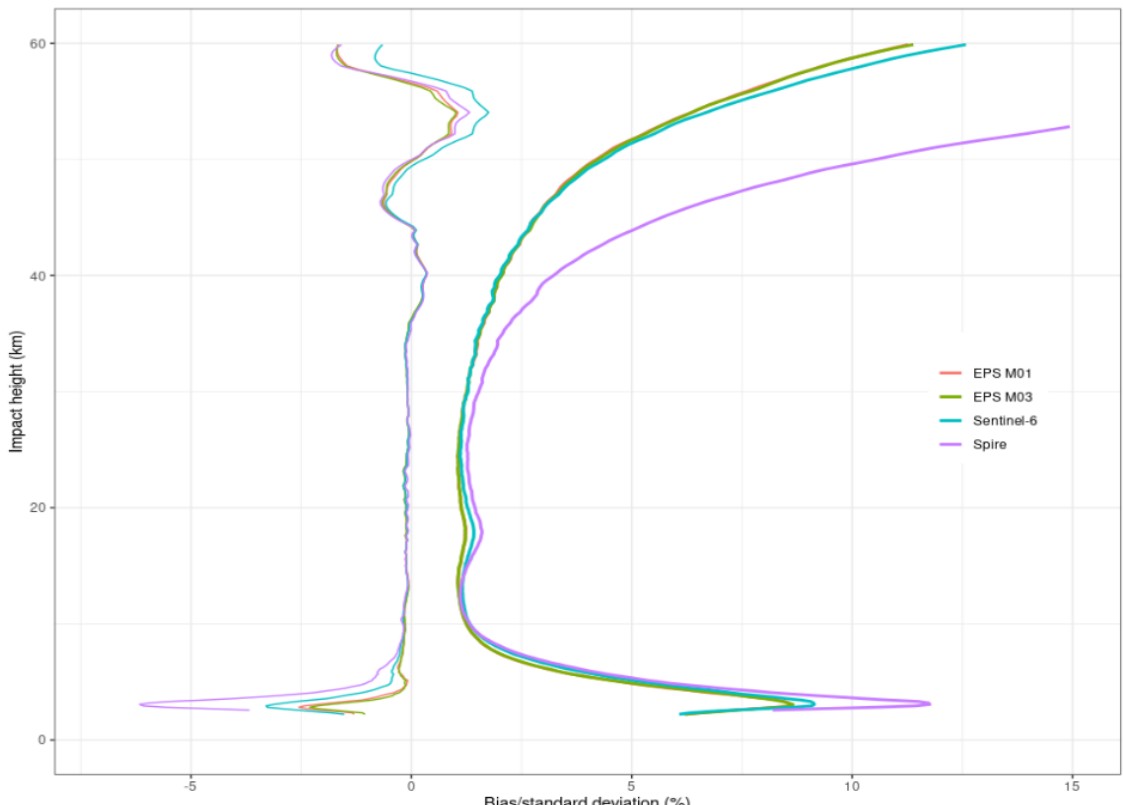

**Figure 16.** Global robust statistics of operational GRAS (M01/B and M03/C), Spire and Sentinel-6A bending angles compared against ECMWF short range forecasts during the analyzed period. Vertical profiles of biases (or systematic deviations) with respective standard (or random) deviation are showed for the different missions.

in the standard deviation for GLONASS occultations compared to GPS occultations, illustrating the specific benefits of the navigation bits removal algorithm's bug fix.

It's noteworthy to say that the GRAS bending angles retrieval algorithm has not yet implemented the SNR-based signal cutoff function (discussed in section 2.1.3), unlike the reprocessed Sentinel-6 and Spire processors. Furthermore, the impact of the L2 cutoff algorithm (referenced in section 2.1.4) is not prominently evident in robust statistics, which are designed to minimize the influence of outliers. Its effects are more discernible in standard statistical analyses that account for the full range of data, including outlier contributions. Consequently, the robust statistics for GRAS depicted in Figures 16 and 17 appear

nearly identical, underscoring the need to consider both robust and standard statistical methods to fully capture the effects of the data processing algorithms on RO data quality.

The most significant difference between figures 16 and 17 is observed in the lower troposphere, below 7 km impact height. In this region, the diverse strategies for signal cutoff and extrapolation, as described in Sokolovskiy et al. (2010) and Culverwell and Healy (2015), give their greatest influence. Notably, the substantial negative biases previously seen in the operational



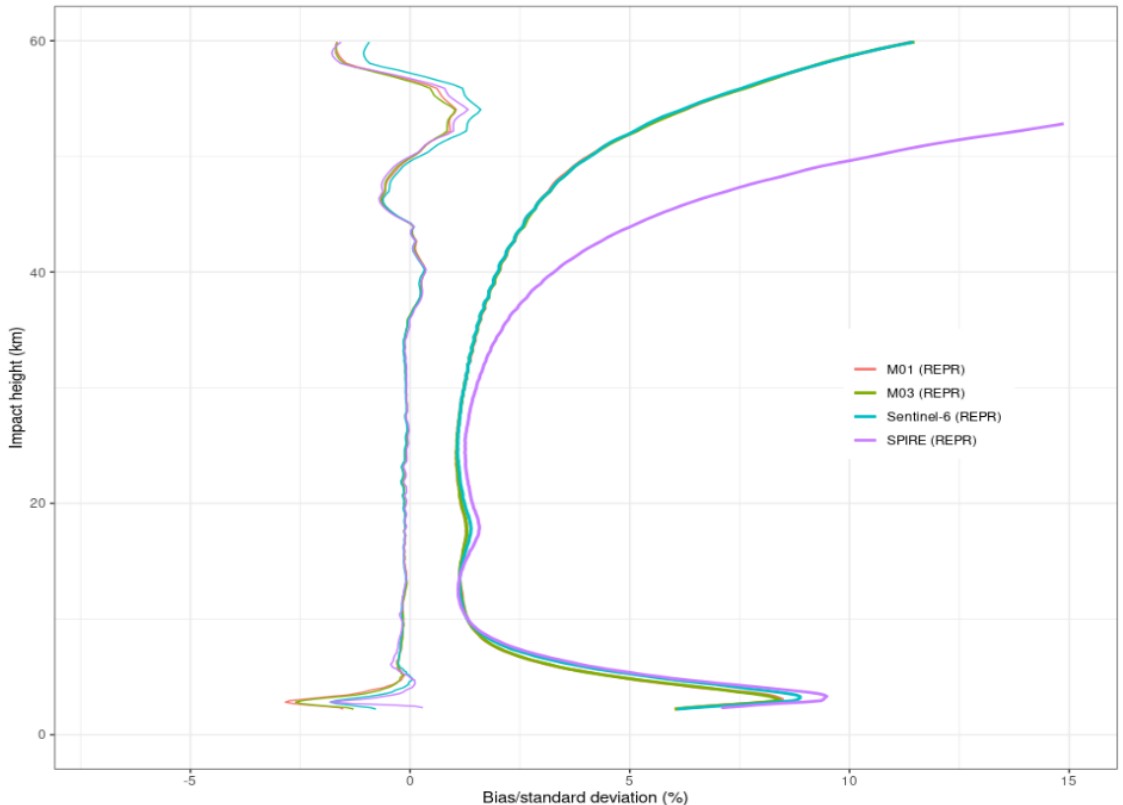

**Figure 17.** Global robust statistics of reprocessed GRAS (M01/B and M03/C), Spire and Sentinel-6A bending angles compared against ECMWF short range forecasts during the analyzed period. Vertical profiles of biases (or systematic deviations) with respective standard (or random) deviation are showed for the different missions.

data from Spire and Sentinel-6A are considerably diminished in the reprocessed data, even surpassing the performance of GRAS B/C data in this region. Figure 19 offers a detailed view of the lower troposphere statistics for both the operational and reprocessed Sentinel-6A data. Beyond the noticeable adjustment in bias structure, the adoption of new cutoff strategies has enabled deeper tropospheric penetration.

The research presented in Sokolovskiy et al. (2010) receives further confirmation from Figure 20, which illustrates how the

positive bias in bending angle retrievals primarily depends on the employment and implementation of a signal cut-off algorithm based on SNR, with the cut-off threshold influencing the tropospheric bias. The systematic error biasing effect is less noticeable at higher latitudes, where the troposphere is drier and more stable, thus experiencing less multi-path interference and reduced instances of super refraction. In contrast, at mid-latitudes and especially in the tropics, where the troposphere is thicker and contains more moisture, the influence of signal cut-off becomes more evident, affecting the bending angle measurements down

to altitudes below 7 km.





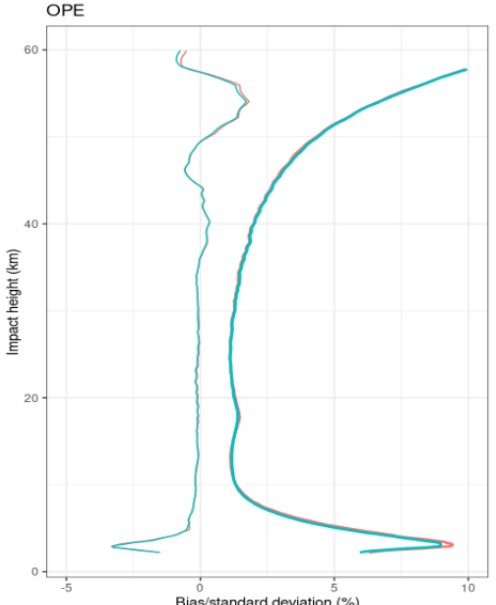 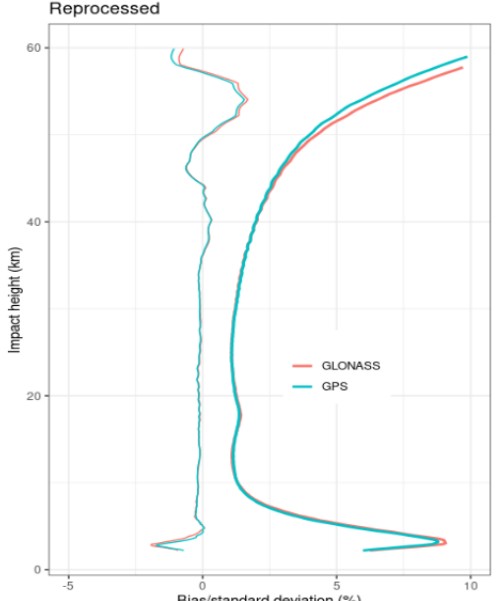

**Figure 18.** Global robust statistics of operational (left) and reprocessed (right) Sentinel-6A bending angles compared against ECMWF short range forecasts during the analyzed period. Vertical profiles of biases (or systematic deviations) with respective standard (or random) deviation are showed split by constellation.

## 4 Conclusions

The Sentinel-6A Michael Freilich satellite, launched into Low Earth Orbit on November 21st, was primarily tasked with continuing the legacy of the altimetry Sentinel mission series. In addition to its main instrument, the altimeter, it is equipped with a GNSS RO TRIG receiver. Since its activation, the RO instrument has consistently provided a significant volume of high-
quality occultation profiles. These profiles, which include data from both GPS and GLONASS satellites, rising and setting occultations, have exceeded the mission's performance targets.

This study utilized a dataset from the last four months of 2021 to evaluate the ability of the Sentinel-6A RO processor version 4.0 to provide high quality bending angle profiles. This version is scheduled to be deployed in operation environment the second quarter of 2024.

The Sentinel-6A RO receiver tracked signals analysis reveals that GPS signals exhibit relatively stable SNR values across different satellites, indicating consistent tracking capabilities for both L1 and L2 GPS signals. In contrast, GLONASS signals show more variability in SNR, with certain satellites exhibiting lower SNR values, especially on the L1 frequency. Further examination of daily averaged SNR values highlighted the effects of satellite maneuvers, such as yaw flips, on tracking capabilities. The analysis also underscores the variability and sensitivity of GLONASS signal tracking to orbital geometries,
contrasting with the more stable performance of GPS signals. Together with SRN, phase noise was also taken into account.

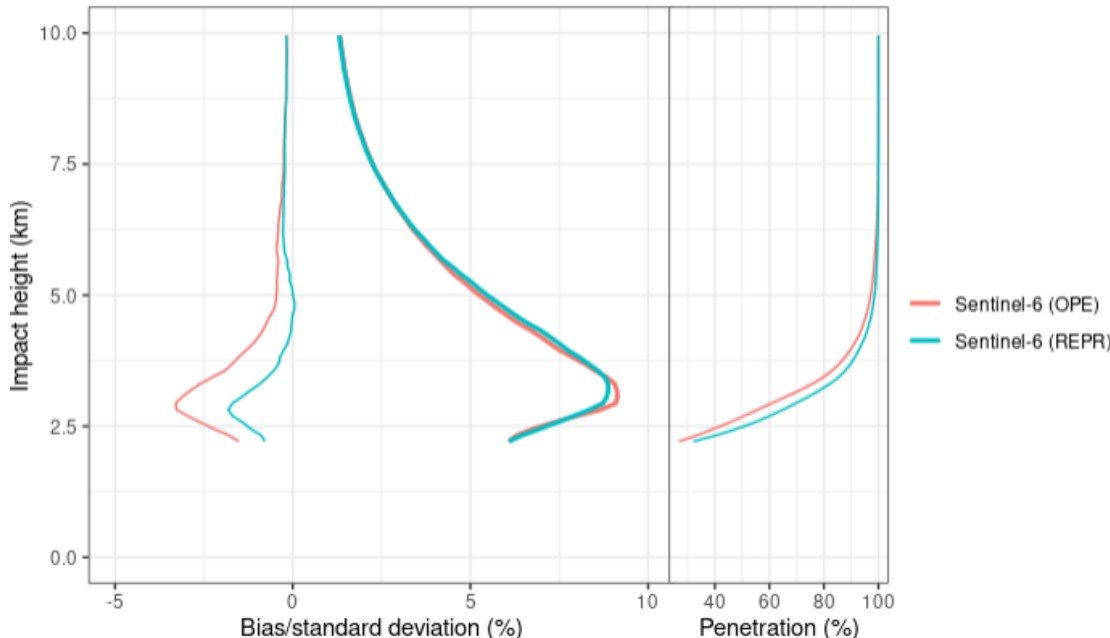

**Figure 19.** Global robust statistics of tropospheric operational and reprocessed Sentinel-6A bending angles compared against ECMWF short range forecasts during the analyzed period. Vertical profiles of biases (or systematic deviations) with respective standard (or random) deviation are showed.

Phase noise can distort the received signal, impacting the accuracy of atmospheric measurements. The analysis indicates that GLONASS occultations are tracked with lower phase noise compared to GPS, affecting the retrieval of bending angles in the upper troposphere.

Investigations on RO data conducted post-launch, during the commissioning phase, addressed the issue of excessively noisy L2 frequency data in some tropospheric occultations. The procedure for extrapolating the L2 signal at lower altitudes using an ionospheric model based on a Chapman Layer to eliminate the ionospheric influence before merging the L1 and L2 bending angle profiles was improved. The determination of the L2 cutoff point was critical to enhance the accuracy and reliability of tropospheric data that accommodates varying solar activity conditions. An initial fixed threshold of $50\mu$rad for the L2 signal cutoff, based on the L1-L2 bending angle difference, showed improvements in bending angle processing. An alternative

approach consisting on integrating the L2 cutoff into the bending angles ionospheric correction process was used. Actual data were robustly fitted into the ionospheric model in order to determine the reference for the L2 signal cutoff algorithm. This new approach was shown to bring substantial improvements into the random errors of the presented statistics.

    The determination of the optimal cut-off points for L1 and L2 GNSS signals based on their SNR levels is a key point for processing data recorded by Open-Loop RO receivers. Different cutoff heights introduce variations in tropospheric bias

structure when RO profiles are compared against ECMWF data. The EUMETSAT RO NTC processor employs the SNR-based





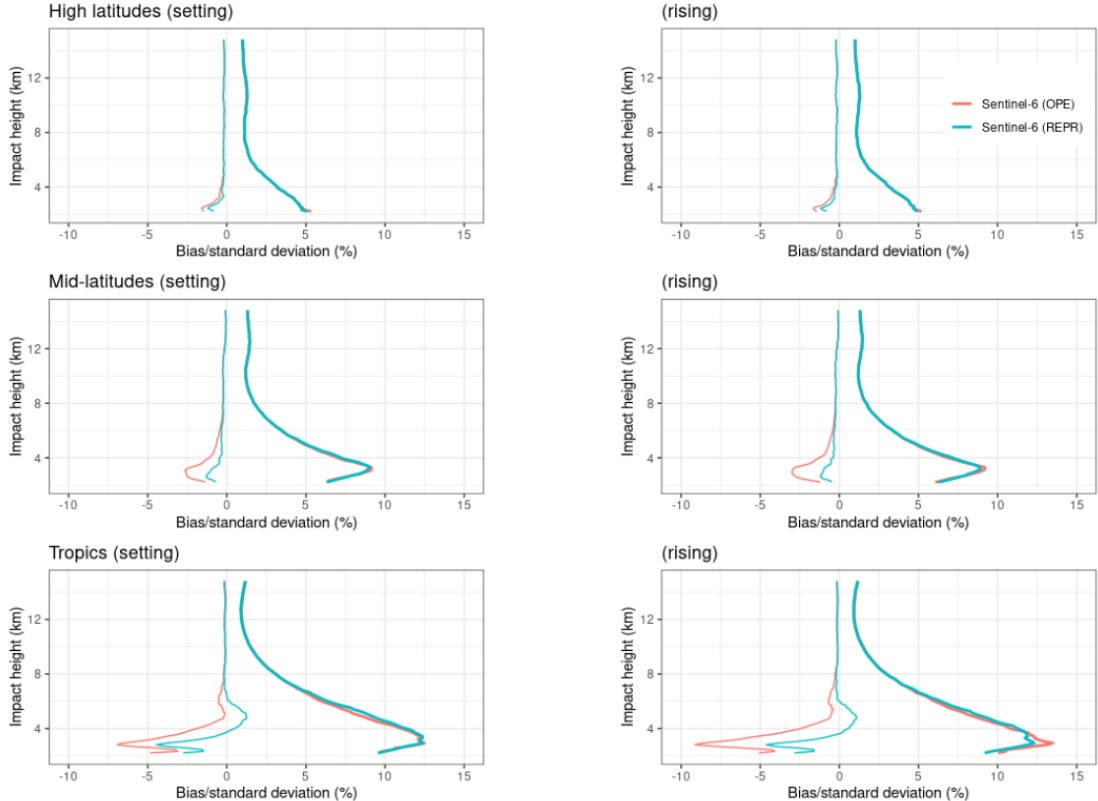

**Figure 20.** Global robust statistics of tropospheric operational and reprocessed Sentinel-6A bending angles compared against ECMWF short range forecasts during the analyzed period. Vertical profiles of biases (or systematic deviations) with respective standard (or random) deviation are showed. plots are stratified by latitudinal bands and split by rising and setting. High latitudes $(60 \deg, 90 \deg \text{ N/S})$, Mid-latitudes $(30 \deg, 60 \deg \text{ N/S})$, Tropics $(-30 \deg, 30 \deg)$.

signal cutoff algorithm as detailed in Sokolovskiy et al. (2010), with minor modifications to address the zero amplitude/SNR occurrences regularly noted in L2 data from JPL receivers, leading to have a tropospheric bias shift of about 1%.

The improvements made to the Sentinel-6A RO-NTC processor for effectively removing navigation bits from the received L1/L2 GNSS signals are discussed. The enhancements have resulted in better-quality bending angle profiles, as demon-

strated by comparisons with ECMWF models, showing improved performance in the lower stratosphere and troposphere for GLONASS signals and across all altitudes for GPS signals. This has contributed to an overall improvement in the data quality produced by the Sentinel-6 RO NTC processor.

Interference signals on the I and Q components of the L2 frequency, not related to GNSS navigation bits, were discussed. Interference at 40/20 Hz was specifically noted in the SNR spectrum of the L2 signals. This phenomenon is more frequent

when the satellite flies over the boundary between Russia and America, suggesting a possible geographic localization of the


interference sources. Although such interference could potentially affect the data accuracy, the Sentinel-6 observations indicate that these signals did not significantly affect the receiver's performance.

Operational and reprocessed Sentinel-6A bending angles profiles were compared against ECMWF short-range forecasts. Operational and Reprocessed GRAS B/C and Spire RO profiles were also used for comparisons purposes. A general agreement

between Sentinel-6A bending angles profiles and ECMWF profiles, similar to that between GRAS and ECMWF, is reported. This alignment underscores the ability of the EUMETSAT RO processors in maintaining consistent and high-quality data across different missions. The comparison also highlighted enhancements in the reprocessed data, which showed reduced random errors and improved adjustments in tropospheric biases. These improvements were primarily attributed to the refinements in the navigation bits removal algorithm and the implementation of an efficient signal cut-off strategy.

*Competing interests.* The contact author has declared that none of the authors has any competing interests



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
