# Peer review of "Assessment of Operational Non-Time Critical Sentinel-6A Michael Freilich Radio Occultation Data: Insights into Tropospheric GNSS Signal Cutoff Strategies and Processor Improvements"

_Atmospheric Measurement Techniques, 2024_

## Author Response (AR1)

**Question:**

- "Availability rate of 99.9%": clarification

Page 1, line 5 states: "This analysis confirms the satellite's

capability to exceed its target of 770 quality checked bending angle

profiles per day, with an availability rate of 99.9%, ..."

This number is repeated in line 70, and it appears that its meaning

should be clarified. Table 1 gives a list of data gaps, and in view of

figure 3 it should be clear that this cannot mean that 99.9% of the days

the mission requirement of 770 occ/day is reached. If that number

refers to conditions where the instrument is working nominally, the data

downlink works, etc., it should be stated as such.

The text also mentions "autonomous or commanded instrument reboots" but

does not give a frequency estimate.

In Line 220 a total data loss of around 1.5% is quoted. Assuming a

roughly constant number of measurements per time, I would like to

understand how these numbers relate.

**Response:**

Line 5: phrases reworked to state that 99.9% availability is reached during full operational periods.

Line 70: phrases reworked to emphasize that the analysed period includes data in commissioning phase and early operational phase, during while several instrument activities were performed. This was done to achieve the mission target of currently provided 770 quality-checked bending angle profiles per day with a 99.9% availability rate.

Line 157: mentioned that the period under study includes commissioning and early operational phase.

Line 187: mentioned that the reported activities are performed during the commissioning and early operational phase.

Line 215: Better described instrument or processor reboots and indicated the current reboots frequency.

Line 217 to 225: Removed some text lines because they are a leftover from previous investigations now outdated. Indeed the text also refers to June 10th 2021 data, not included in the analysed period of this paper. Sorry for that.

\*\*\*\*\*

**Question:**

- Line 127: formulation

"The Sentinel-6A RO instrument's performance, is required to provide ..."

I think the following reads better:

"The Sentinel-6A RO instrument requirement is to provide ..."

**Response:**

Line 127: Fixed, thank you

\*\*\*\*\*\*

**Question:**

- Line 168: clarification

"For both datasets, bending angle profiles were thinned to 247 vertical

levels"

I assume that the bending angle profiles were thinned to the common set

of 247 vertical levels as for near-real time disseminated BUFR products

as used by NWP.

**Response:**

Line 168: Yer correct, text updated adding the info you report. Thank you.

\*\*\*\*\*

**Question:**

- Line 173: clarification

"This forward-modelling process converts ECMWF data on temperature,

humidity, pressure, and geopotential, provided at 137 vertical levels at

the reference occultation position, to 247 bending angle levels."

My understanding is that ECMWF provides temperature, humidity, \*surface\*

pressure, and \*surface\* geopotential (orography). The forward-modelling

process (e.g. Abel integral) is not simply a conversion; it \*derives\*

the model-equivalent of the bending angles at the selected 247 levels

from the input variables. Please try to find an adequated formulation.

**Response:**

Line 173: Sentences reformulated in order to make the description of what forward modelling is more adequate. Thank you.

\*\*\*\*\*

**Question:**

- Line 185: "770 bending angle profile" - use plural: profiles

**Response:**

- Line 185: Fixed thank you.

\*\*\*\*\*

**Question:**

- Line 200: "Subsequent Sentinel-6A RO processor updates made it more

robust to these kind of issues." - either "this kind" or "these kinds"

**Response:**

- Line 200: Fixed thank you.

\*\*\*\*\*

**Question:**

- Line 229: "or RO profiles" - "or" should read "of"

**Response:**

- Line 229: Fixed thank you

\*\*\*\*\*

**Question:**

- Line 230: clarification

"It's worth nothing that lower SNR missions like SPIRE have demonstrated

the ability to systematically detect key atmospheric features."

It is certainly a matter of taste what "key atmospheric features" are.

If the authors mean (lower) tropospheric/atmospheric features, they

might say so.

**Response:**

- Line 230: I meant actually "critical atmospheric features, particularly in the lower troposphere". Text updated and also cited a paper where SPIRE occultations are assessed. Thank you.

\*\*\*\*\*

**Question:**

- Figure 6 and text after line 255: clarification needed

Not being an expert, I fail to understand yaw flips decrease the SNR

both for rising and for setting occultations even for \*GPS\* signals.

Figure 5 top row suggests that SNR for GPS is roughly independent of

PRN, so that a naive reader might guess that exchanging the antennae

would decrease the SNR for setting but increase the SNR for rising.

So forgetting GLONASS for the argument, why do we see what figure 6 top

row shows? I cannot understand it from the text in lines 255ff.

**Response:**

- Line 255: Sorry you are right; the involvement of the beam forming is not that clear. I´ve rearranged the sentences like this: The time series in this figure illustrates the impact of yaw flips executed in early September and November, particularly when not followed by beam forming adjustments. In such cases, the beam forming that was initially optimized for the rising and setting antennas becomes misaligned once the antennas are switched, leading to a significant reduction in SNR.

\*\*\*\*\*

**Question:**

- Figure 8: I only see "bias" in the plot, so "standard deviation" should

be removed from the x-axis label.

Furthermore, in the caption: "are showed" should read "are shown".

This should also be fixed in more places (fig.9,11,13,16,17,18,19,20).

**Response:**

- Figure 8,9,11,13,16,17,18,19,20 all fixed. Thank you.

\*\*\*\*\*

**Question:**

- Line 369: "the JPL provide navigation bit data file" -> \*provided\*

**Response:**

- Line 369: Fixed. Thank you.

\*\*\*\*\*

**Question:**

- Line 408: "where other error contribution, in particular." ->

\*contributions\*.

**Response:**

- Line 408: Fixed. Thank you

**\*\*\*\**

**Question:**

- Figure 14, caption: "occultation recorder" -> \*recorded\*

**Response:**

- Figure 14: Fixed. Thank you

**\*\*\*\*\**

**Question:**

- Line 460: "SRN" -> "SNR"

**Response:**

- Line 460: Fixed. Thank you.

\*\*\*\*\*

**Question:**

- Line 469: formulation

"An alternative approach consisting on integrating the L2 cutoff ..."

Maybe better:

"An alternative approach of integrating the L2 cutoff ..."

**Response:**

Line 469: Fixed. Thank you

\*\*\*\*\*

**Question:**

- Line 477: "leading to have a tropospheric bias shift" - remove "have"?

**Response:**

- Line 477: Fixed, thank you.

\*\*\*\*\*

**Question:**

- Line 490: "bending angles profiles" -> "bending angle profiles"

**Response:**

- Line 490: Fixed. Thank you

\*\*\*\*\*

**Question:**

- References, line 563 (and citations in the text):

Author "Ho, Shu-Peng" appears as "peng Ho". Not sure if this is a

problem with the bibliography style or the way the reference is handled.

The actually used citation looks strange to me, and a fix should be

attempted.

**Response:**

- References, line 563 (and citations in the text): Fixed, Thank you.

\*\*\*\*\*

**Question:**

-page 5, l. 126: delete "then"

-page 5, l. 132: use "emphasize" instead of "underscores"

-page 7, In caption Table 1remove ")"

**Response:**

All fixed, thank you.

\*\*\*\*\*

**Question:**

-page 8, l. 173: Which option in the forward operator is used? 1D or 2D?

**Response:**

Added 1D in the text. Thank you

\*\*\*\*\*

**Question:**

-page 8, l. 177 and "which" after "phase, "

-page 8, l. 184 add "number" after "combined total"

**Response:**

All fixed, thank you

\*\*\*\*\*

**Question:**

-page 10 l.237: "PRN" Please write out acronym when used for the first time.

**Response:**

Actually I have removed the usage of PRN because the PRN more identifies GPS satellites than GLONASS ones. So I moved to using the more general "satellite number" instead of PRN. Do you agree? Thank you.

\*\*\*\*\*\*

**Question:**

-page11, in caption of Figure 4: "SLTA" This is explained later on page 14 but maybe best to do it here, when used for the first time.

**Response:**

Explained the first time it is mentioned. thank you

\*\*\*\*\*

**Question:**

-page 11, l.258: "nothing" should be "noting"

**Response:**

Fixed, thank you

\*\*\*\*\*

**Question:**

-page 12, Fig.5: Add an x – axis, e.g "Various satellites". Also, legend is difficult to read (too small).

**Response**:**

Ok plot updated. Legend removed and satellites names moved directly to the X-Axis. It looks much readable now. Thank you.

\*\*\*\*\*

**Question:**

-page 12, Fig.6: Please add more details in the caption what can be seen in the different panels. Especially for top right and bottom left it is difficult to see the difference between them (at least from the title).

**Response:**

Do you mean Figure 6 in page 13? in case so I have updated the caption this way: Daily average mean SNRs measured between 60 and 80 km SLTA, categorized by GNSS signal code and split between rising and setting occultations. The top left panel shows values for GPS and GLONASS L1C codes, while the top right, bottom left, and bottom right panels report SNR values for GPS L2C, GPS L2P, and GLONASS L2CA codes, respectively. Titles are removed but legends now better indicate the content of the plots. Thank you

\*\*\*\*\*

**Question:**

-page 17, Fig.9: Define the different lines. Std dev in bold and bias are thin lines?

**Response**:**

You are right. Fixed for the plot you indicate but also for the other plots where the biases and standard deviations are shown. Thank you.

**\*\*\*\**

**Question:**

-page 18, l. 370: "these" should be "this"

-page 21, Fig.13. "bite" should be "bit"

**Response:**

All fixed thank you